# Discovery of widespread transcription initiation at microsatellites predictable by sequence-based deep neural network

Mathys Grapotte[1,2,3,211], Manu Saraswat[1,2,211], Chloé Bessière[1,2,211], Christophe Menichelli [1,4], Jordan A. Ramilowski [5], Jessica Severin [5], Yoshihide Hayashizaki [6], Masayoshi Itoh [6], Michihira Tagami[5], Mitsuyoshi Murata[5], Miki Kojima-Ishiyama[5], Shohei Noma[5], Shuhei Noguchi[5], Takeya Kasukawa [5], Akira Hasegawa[5], Harukazu Suzuki [5], Hiromi Nishiyori-Sueki[5], Martin C. Frith[7,8,9], FANTOM consortium*, Clément Chatelain[3], Piero Carninci [5], Michiel J. L. de Hoon [5], Wyeth W. Wasserman [10], Laurent Bréhélin [1,4 ✉] & Charles-Henri Lecellier [1,2,4 ✉]

Using the Cap Analysis of Gene Expression (CAGE) technology, the FANTOM5 consortium provided one of the most comprehensive maps of transcription start sites (TSSs) in several species. Strikingly, ~72% of them could not be assigned to a specific gene and initiate at unconventional regions, outside promoters or enhancers. Here, we probe these unassigned TSSs and show that, in all species studied, a significant fraction of CAGE peaks initiate at microsatellites, also called short tandem repeats (STRs). To confirm this transcription, we develop Cap Trap RNA-seq, a technology which combines cap trapping and long read MinION sequencing. We train sequence-based deep learning models able to predict CAGE signal at STRs with high accuracy. These models unveil the importance of STR surrounding sequences not only to distinguish STR classes, but also to predict the level of transcription initiation. Importantly, genetic variants linked to human diseases are preferentially found at STRs with high transcription initiation level, supporting the biological and clinical relevance of transcription initiation at STRs. Together, our results extend the repertoire of non-coding transcription associated with DNA tandem repeats and complexify STR polymorphism.

[1] Institut de Biologie Computationnelle, Montpellier, France. [2] Institut de Génétique Moléculaire de Montpellier, University of Montpellier, CNRS, Montpellier, France. [3] SANOFI R&D, Translational Sciences, Chilly Mazarin, France. [4] LIRMM, Univ Montpellier, CNRS, Montpellier, France. [5] RIKEN Center for Integrative Medical Sciences, Yokohama, Kanagawa, Japan. [6] RIKEN Preventive Medicine and Diagnosis Innovation Program, Wako, Saitama, Japan. [7] Artificial Intelligence Research Center, AIST, Tokyo, Japan. [8] Graduate School of Frontier Sciences, University of Tokyo, Chiba, Japan. [9] AIST-Waseda University CBBD-OIL, AIST, Tokyo, Japan. [10] Centre for Molecular Medicine and Therapeutics at the Child and Family Research Institute, Department of Medical Genetics, University of British Columbia, Vancouver, BC, Canada. [211] These authors contributed equally: Mathys Grapotte, Manu Saraswat, Chloé Bessière. *A list of authors and their affiliations appears at the end of the paper. ✉email: brehelin@lirmm.fr; charles.lecellier@igmm.cnrs.fr

R NA polymerase II (RNAPII) transcribes many loci outside annotated protein-coding gene promoters[1,2] to generate a diversity of RNAs, including for instance enhancer RNAs[3] and long noncoding RNAs (lncRNAs)[4]. In fact, >70% of all nucleotides are thought to be transcribed at some point[1,5,6]. Using the Cap Analysis of Gene Expression (CAGE) technology[7,8], the FANTOM5 consortium provided one of the most comprehensive maps of TSSs in several species[2]. Integrating multiple collections of transcript models with FANTOM CAGE datasets, Hon et al. built a new annotation of the human genome (FANTOM CAGE-Associated Transcriptome, FANTOM CAT), with an atlas of 27,919 human lncRNAs, among them 19,175 potentially functional RNAs[4]. Despite this annotation, many CAGE peaks remain unassigned to a specific gene and/or initiate at unconventional regions, outside promoters or enhancers, providing an unprecedented mean to further characterize noncoding transcription within the genome "dark matter"[9] and to decode part of the transcriptional "noise".

Noncoding transcription is indeed far from being fully understood[10] and some authors suggest that many of these transcripts, often faintly expressed, can simply be "noise" or "junk"[11,12]. On the other hand, many non annotated RNAPII transcribed regions correspond to open chromatin[1] and cis-regulatory modules bound by transcription factors (TFs)[13]. Besides, genome-wide association studies showed that trait-associated loci, including those linked to human diseases, can be found outside canonical gene regions[14–16]. Together, these findings suggest that the noncoding regions of the human genome harbor a plethora of potentially transcribed functional elements, which can drastically impact genome regulations and functions[9,16].

The human genome is scattered with repetitive sequences, and a large portion of noncoding RNAs derives from repetitive elements[17,18], in particular DNA tandem repeats, such as satellite DNAs[19] and minisatellites[20]. Microsatellites, also called short tandem repeats (STRs), constitute the third class of DNA tandem repeats. They correspond to repeated DNA motifs of 2–6 bp and constitute one of the most polymorphic and abundant repetitive elements[21]. Classes of STRs can be defined based on the repeated DNA motif (e.g., $(AC)_n$ will correspond to all STRs with repeats of the dinucleotide AC). STR polymorphism, which corresponds to variation in the number of repeated DNA motif (i.e., STR length), is presumably due to their susceptibility to slippage events during DNA replication. STRs have been shown to widely impact gene expression and to contribute to expression variation[22–25]. Some constitute genuine expression Quantitative Trait Loci (eQTLs)[23,24], called eSTRs[23]. At the molecular level, STRs can for instance affect expression by inducing inhibitory DNA structures[26] and/or by modulating TF binding[27,28].

Provided the abundance of STRs on the one hand and the widespread transcription of the genome, including at repeated elements, on the other hand, we hypothesize that transcription initiation also occurs at STRs. To test this hypothesis, we probe CAGE data collected by the FANTOM5 consortium[2] using the STRs catalog built by Willems et al.[29]. We specifically show that a significant portion of CAGE peaks (~8.6%) initiate at STRs. This transcription is confirmed by Cap Trap RNA-seq (CTR-seq), a technology that combines cap trapping and long-read MinION sequencing. Transcription of STR-containing RNAs has previously been reported in several species[30–33]. We report here that thousands of STRs can also initiate transcription in human and mouse, therefore not being only a mere passenger in other RNAs but containing genuine TSSs. We further learn sequence-based Convolutional Neural Networks (CNNs) able to predict these transcription initiation levels with high accuracy (correlation between observed and predicted CAGE signal >0.65 for 14 STR classes with >5000 elements). These models unveil the importance of STR flanking sequences in distinguishing STR classes, one from the other, and also in predicting transcription initiation. We finally show that genetic variants linked to human diseases, are located, not only within, but also around STRs associated with high transcription initiation levels.

## Results

**CAGE peaks are detected at STRs.** We first intersected the coordinates of 1,048,124 CAGE peak summits[2] with that of 1,620,030 STRs called by HipSTR[29]. We found that 89,948 CAGE peaks (~8.6%) initiate at 84,555 STRs (Fig. 1a and Supplementary Fig. 1). As a comparison, only 2.3% of an equal number of randomly selected intervals with equivalent size intersected with CAGE peaks (Fisher's exact test P value < 2.2e-16). Among CAGE peaks intersecting with STRs, 10,727 correspond to TSSs of FANTOM CAT transcripts[4] and 8823 to enhancer boundaries[3] (Supplementary Data 1). Note that the FANTOM CAT annotation was shown to be more accurate in 5' end transcript definitions compared to other catalogs (GENCODE[34], Human BodyMap[35], and miTranscriptome[36]), because transcript models combine various independent sources (GENCODE release 19, Human BodyMap 2.0, miTranscriptome, ENCODE and an RNA-seq assembly from 70 FANTOM5 samples) and FANTOM CAT TSSs were validated with Roadmap Epigenome DHS and RAM-PAGE datasets[4]. This transcription does not correspond to random noise because the fraction of STRs harboring a CAGE peak within each class differs depending on the STR class, without any link with their abundance (Fig. 1a, c). Some STR classes with low abundance are indeed more often associated with a CAGE peak than more abundant STRs (Fig. 1a, c, compare for instance $(CTTTTT)_n$ or $(AAAAG)_n$ vs. $(AT)_n$ or $(ATTT)_n$). Likewise, the number of STRs associated with CAGE peaks cannot merely be explained by their length, as several STR classes have similar length distribution but very different fractions of CAGE-associated loci (compare for instance $(AT)_n$ and $(GT)_n$ in Fig. 1c and Supplementary Fig. 2).

We computed the tag count sum along each STR ± 5 bp, and averaged the signal across 988 FANTOM5 libraries. We noticed the existence of very low (tag count = 1) CAGE counts along STRs, which artificially increase the signal (see examples in Fig. 1a, Spearman correlation coefficient between sum CAGE tag count along STR and STR length ~0.26). To remove any dependence between STR length and CAGE signal, the mean tag count was normalized by the length of the window used to compute the signal (i.e., STR length + 10 bp). Looking directly at this CAGE signal (not CAGE peaks) along the genome, we observed that some STR classes are more transcribed than others (Fig. 1d, compare $(CGG)_n$ or $(CCG)_n$ vs. $(AAGG)_n$ or $(AAAAT)_n$). No drastic difference in terms of CAGE signal was noticed between intra- and intergenic STRs (Supplementary Fig. 3). Looking at each STR class separately, we confirmed that our CAGE signal computation is not sensitive to the STR length (Supplementary Fig. 4). Supplementary Fig. 4 also shows that STRs with different lengths can be associated with the same CAGE signal while, conversely, two STRs with different CAGE signals can have the same length. Thus, considering transcription, STR polymorphism appears to not only rely on their length (number of repeated elements). Transcription initiation, therefore, appears to complexify STR polymorphism.

**CAGE tags correspond to genuine transcriptional products.** CAGE read detection at STRs faces two problems. First, CAGE tags can capture not only TSSs but also the 5' ends of post-transcriptionally processed RNAs[37]. To clarify this point, we used a strategy described by de Rie et al.[38], which compares CAGE tags

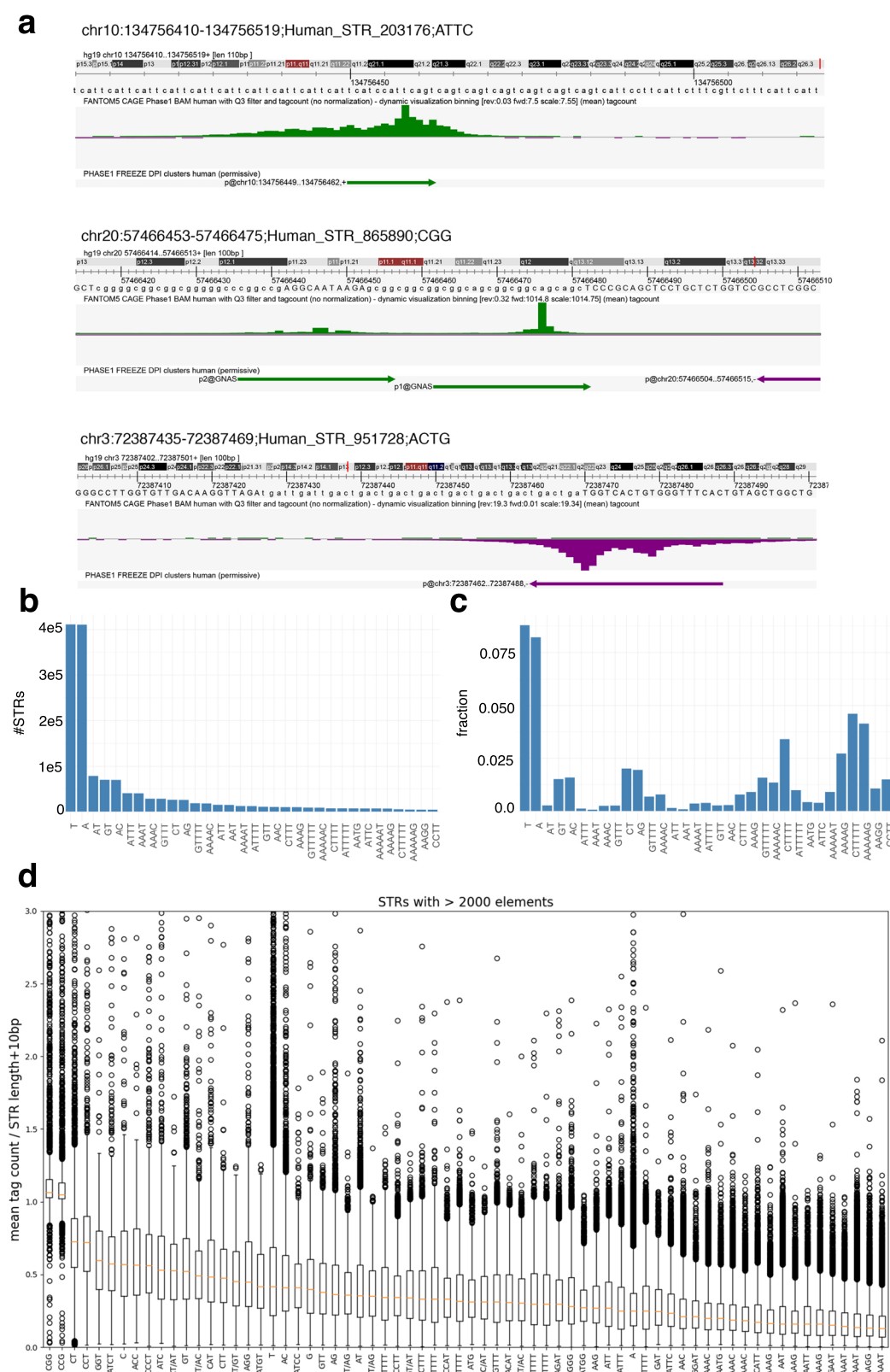

obtained by Illumina (ENCODE) vs. Heliscope (FANTOM) technologies. Briefly, the 7-methylguanosine cap at the 5' end of CAGE tags produced by RNAPII can be recognized as a guanine nucleotide during reverse transcription. This artificially introduces mismatched Gs at Illumina tag 5' end, not detected with Heliscope sequencing, because it skips the first nucleotide[38]. We

then evaluated the existence of this G bias in CAGE tags corresponding to peaks detected at STRs, peaks assigned to genes (for positive control), and peaks intersecting the 3' end of precursor microRNAs (pre-miRNAs for a negative control) (Fig. 2). While most CAGE tag 5' ends perfectly match the sequences of pre-miRNA 3'end in all cell types tested, as previously reported[38], a G

**Fig. 1 CAGE peaks are detected at STRs. a** Three examples of STRs associated with a CAGE peak. The Zenbu browser[79] was used. top track, hg19 genome sequence; middle track, CAGE tag count as mean across 988 libraries (BAM files with Q3 filter were used); bottom track, CAGE peaks as called in ref. [2]. **b** Number of STRs per STR class. For sake of clarity, only STR classes with >2000 loci are shown. **c** Fraction of STRs associated with a CAGE peak in all STR classes considered in **b**. **d** CAGE signal at STR classes with >2000 loci. CAGE signal was computed as the mean raw tag count of each STR (tag count in STR ± 5 bp) across all 988 FANTOM5 libraries. This tag count was further normalized by the length of the window used to compute the signal (i.e., STR length + 10 bp). The orange bar corresponds to the median value. The lower and upper hinges correspond to the first and third quartiles (the 25th and 75th percentiles). The upper whisker extends from the hinge to the largest value no further than 1.5 × IQR from the hinge (where IQR is the interquartile range or distance between the first and third quartiles). The lower whisker extends from the hinge to the smallest value at most 1.5 × IQR of the hinge. Data beyond the end of the whiskers are plotted individually.

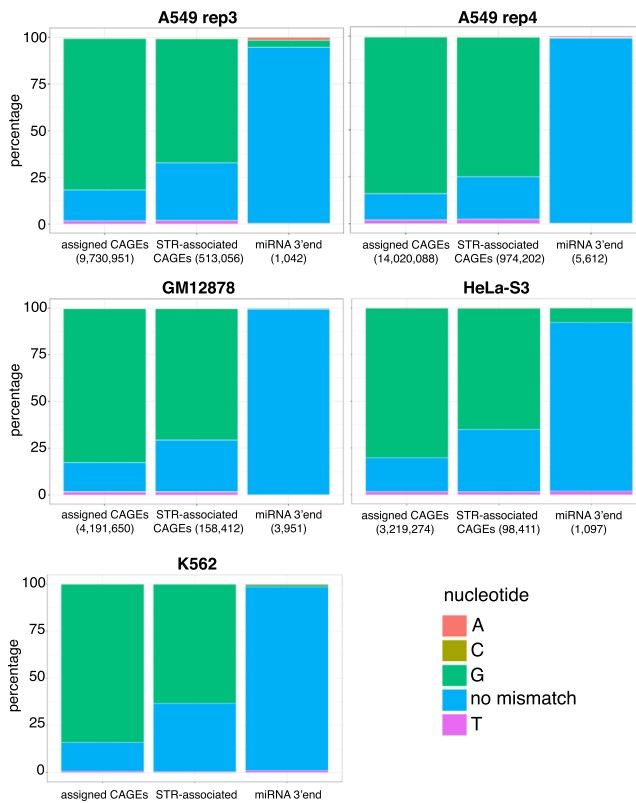

**Fig. 2 CAGE tags initiating at STRs are truly 5'-capped.** G bias in ENCODE CAGE tags (bam files from nuclear fraction, polyA+) was assessed at FANTOM5 CAGE peaks assigned to genes (positive control) and CAGE peaks initiating at STRs. G bias at pre-microRNA 3' ends was also assessed as a negative control. Five libraries were analyzed corresponding to A549 (replicates 3 and 4), GM12878, HeLa-S3, and K562 cells. The number of intersecting tags in each case is indicated in the bracket.

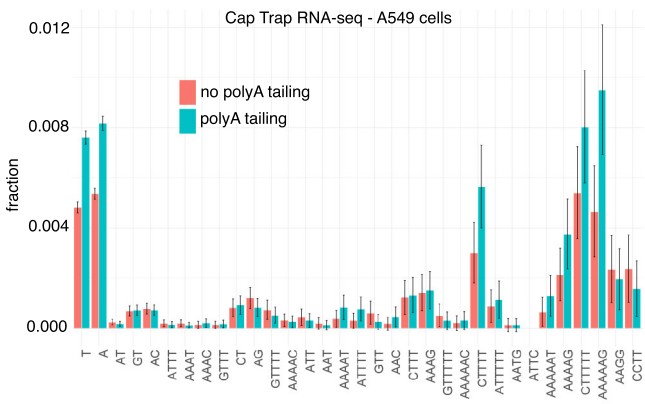

**Fig. 3 CTR-seq confirms the existence of transcription initiation at STRs.** The fractions of STRs associated with at least one CTR-seq long-read start site were computed for all STR classes considered in Fig. 1b. RNAs were collected in A549 cells. Reverse transcription was preceded (blue) or not (red) by polyA tailing. Binomial proportion 95% confidence intervals are indicated and centered on the fraction value (y axis).

expected given the depth of MinION sequencing in only one cell line, the number of STRs associated with long reads is lower than that obtained with CAGE sequencing collected in 988 libraries ($n = 5472$ and 7812, respectively, with and without polyA tailing with 2291 STRs associated with long reads in both libraries). Among these 2291 STRs, 904 (39%) are also associated with a CAGE peak. Thus, compared to the reproducibility of MinION sequencing in both libraries (only 2291 STRs in common out of 5472 (42%) or 7812 (29%)), CAGE and CTR-seq sequencing results are overall in agreement. In fact, STR classes associated with CAGE peaks correspond to those associated with CTR-seq reads (Fig. 3 compared to Fig. 1c). The Spearman correlation $\rho$ between the fractions of STRs associated with CAGE and MinION reads with and without polyA tailing equals 0.88 and 0.89 respectively. Besides, 301 out of 904 STRs associated with both CAGE peak and CTR-seq long read correspond to TSSs of FANTOM CAT transcripts and 54 to enhancer boundaries. Overall, CTR-seq confirms CAGE data and the existence of transcription initiating at STRs. The similarity of the results obtained with and without the polyA tailing step also indicates that RNAs initiating at STRs are mostly polyadenylated.

**Transcription initiation at STRs exhibits specific features**. We further looked at the subcellular localization of STR-initiating transcripts and used CAGE sequencing data generated after cell fractionation (see "Methods" section). While the majority of CAGE tags, including those assigned to genes, are detected in both the nucleus and cytoplasm, CAGE tags initiating at STRs are mostly detected in the nuclear compartment (Fig. 4a). Functionally distinct RNA species were previously categorized by their transcriptional directionality[39]. We then sought to compute the

bias was clearly observed when considering assigned CAGEs and CAGEs detected at STRs, confirming that the vast majority of STR-associated CAGE tags are truly capped. We also confirmed that STRs located within RNAPII-binding sites exhibit a stronger CAGE signal than STRs not associated with RNAPII-binding events (Supplementary Fig. 5).

Second, because of their repetitive nature, mapping CAGE reads to STRs is problematic and may yield ambiguous results. To circumvent this issue, we developed CTR-seq, which combines cap trapping and long-read MinION sequencing. With this technology, the median read length is >500 bp, thereby greatly limiting the chance of erroneous mapping. Two libraries were generated in A549 cells, including or not polyA tailing. This polyA tailing step before reverse transcription allows the detection of polyA-minus noncoding RNAs. Long reads initiating at STRs were readily detected in both libraries (Fig. 3). As

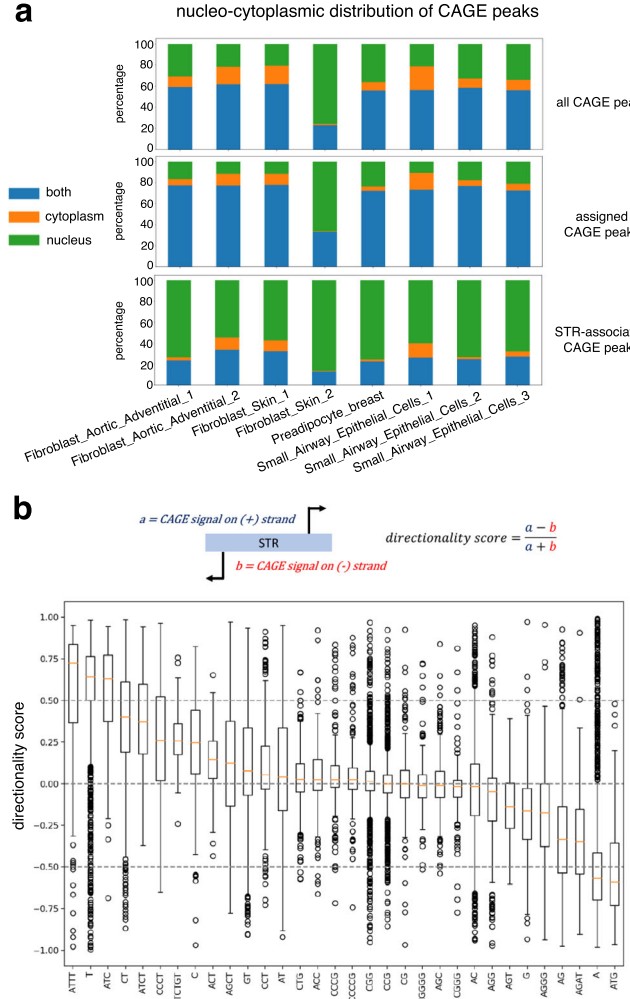

**a**

nucleo-cytoplasmic distribution of CAGE peaks

*(all CAGE peaks / assigned CAGE peaks / STR-associated CAGE peaks — legend: both (blue), cytoplasm (orange), nucleus (green). x-axis libraries: Fibroblast_Aortic_Adventitial_1, Fibroblast_Aortic_Adventitial_2, Fibroblast_Skin_1, Fibroblast_Skin_2, Preadipocyte_breast, Small_Airway_Epithelial_Cells_1, Small_Airway_Epithelial_Cells_2, Small_Airway_Epithelial_Cells_3)*

**b**

$a = $ CAGE signal on (+) strand

STR

$$directionality\ score = \frac{a - b}{a + b}$$

$b = $ CAGE signal on (-) strand

**Fig. 4 CAGE peaks at STRs exhibit specific features. a** STR-associated CAGE tags are preferentially detected in the nuclear compartment. For each indicated library (*x* axis) and each CAGE peak, CAGE expression (TPM) was measured in nuclear and cytoplasmic fractions. Each CAGE peak was then assigned to the nucleus (if only detected in the nucleus), cytoplasm (if only detected in the cytoplasm), or both compartments (if detected in both compartments). The number of CAGE peaks in each class is shown for each sample as a fraction of all detected CAGE peaks. The sample *Fibroblast_Skin_2* likely represents a technical artifact. Analyses were conducted considering 201,802 FANTOM5 CAGE peaks (top), 54,001 CAGE peaks assigned to genes (middle), and 14,509 CAGE peaks associated with STRs (bottom). **b** Boxplots of directionality scores for each STR class with >100 elements. A score of 0 means that the transcription is bidirectional and occurs on both strands. A score of 1 indicates that transcription occurs on the (+) strand, while −1 indicates transcription exclusively on the (−) strand (STRs being defined on the (+) strand in HipSTR catalog). Boxplots are defined as in Fig. 1d.

directionality score, as defined by Hon et al. in ref. [4], for each STR associated with CAGE signal (Fig. 4b). Briefly, this score corresponds to the difference between the CAGE signal on the (+) strand and that on the (−) strand divided by their sum (in HipSTR catalog, STRs are systematically defined on the (+) strand i.e., $(T)_n$ on (−) strand are defined as $(A)_n$). A score equals to 1 or −1 indicates that transcription is strictly oriented toward the (+) or (−) strand, respectively. A score close to 0 indicates that the transcription is balanced and that it occurs equally on the

(+) and (−) strands. As shown in Fig. 4b, some STR classes are associated with directional transcription either on the (+) (e.g., $(ATTT)_n$, $(T)_n$) or (−) (e.g., $(A)_n$, $(ATG)_n$) strand, while others are bidirectional and balanced ($(CGG)_n$, $(CCG)_n$). Furthermore, scores obtained at $(A)_n$ STRs are mostly negative, while scores obtained at $(T)_n$ STRs are mostly positive. This indicates that transcription initiation preferentially occurs on the strand where $(T)_n$ STRs are found. The fact that transcription can be either directional or bidirectional depending on the STR class suggests that transcription initiation at STRs is governed by different features, which are specific to STR classes. We looked for motifs known to be involved in transcription directionality at canonical TSSs, namely, polyadenylation sites (polyA sites) and U1-binding sites[40]. Sequences encompassing −3/+10bp[41] around FANTOM CAT 5' donor splice sites were used to build a position weight matrix (PWM) corresponding to the U1-binding site (Supplementary Fig. 6). This PWM was further used to scan 2 kb-long sequences centered around $(T)_n$ 3' end and FANTOM CAT TSSs (used as positive control). $(T)_n$ STRs have been chosen as a prototype of directional transcription initiation at STRs (Fig. 4b). While we confirmed enrichment of potential U1-binding sites downstream FANTOM CAT TSSs[40], such enrichment was not observed downstream $(T)_n$ 3' ends (Supplementary Fig. 6). Likewise, polyA sites are clearly enriched upstream FANTOM CAT TSSs, but this observation does not hold true for $(T)_n$ STRs (Supplementary Fig. 6). Our results extend the findings of Ibrahim et al., who reported that a single model of transcription initiation within and across eukaryotic species is not evident[42].

**A sequence-based deep learning model reveals that features governing transcription initiation depend on the STR classes.** We further probed transcription initiation at STRs using a machine-learning approach. We used a deep Convolutional Neural Network (CNN), which is able to successfully predict CAGE signal in large regions of the human genome[43,44]. This type of machine-learning approach takes as input the DNA sequence directly, without the need to manually define predictive features before analysis. The first question that arose was then to determine the sequence to use as input.

We first sought to build a model common to all STR classes to predict the CAGE signal as computed in Fig. 1d. Note that, because we used mean signal across CAGE libraries, our model is cell-type agnostic. This choice was motivated by the observation that the CAGE signal at STRs in each library is very sparse, thereby strongly reducing the prediction accuracy of our model. As input, we used sequences spanning 50 bp around the 3' end of each STR. Model architecture and constructions of the different sets used for learning are detailed in the "Methods" section and in Supplementary Fig. 7. Source code is available at https://gite.lirmm.fr/ibc/deepSTR. The accuracy of our model was computed as Spearman correlation between the predicted and the observed CAGE signals on held-out test data (see "Methods"). The performance of this global model was overall high ($P \sim 0.72$), indicating that transcription initiation at STRs can indeed be predicted by sequence-level features. However, looking at the accuracy for each STR class, we noticed drastic differences with accuracies ranging from <0.6 to 0.81 depending on the STR class (Fig. 5a, blue dots). The global model is notably accurate for the most represented STR class (i.e., $(T)_n$ with 766,747 elements), but performs worse in other STR classes. Differences in accuracies are not simply linked to the number of elements available for learning in each STR class. They rather suggest that, as proposed above (Fig. 4b), transcription initiation may be governed by features specific to each STR class.

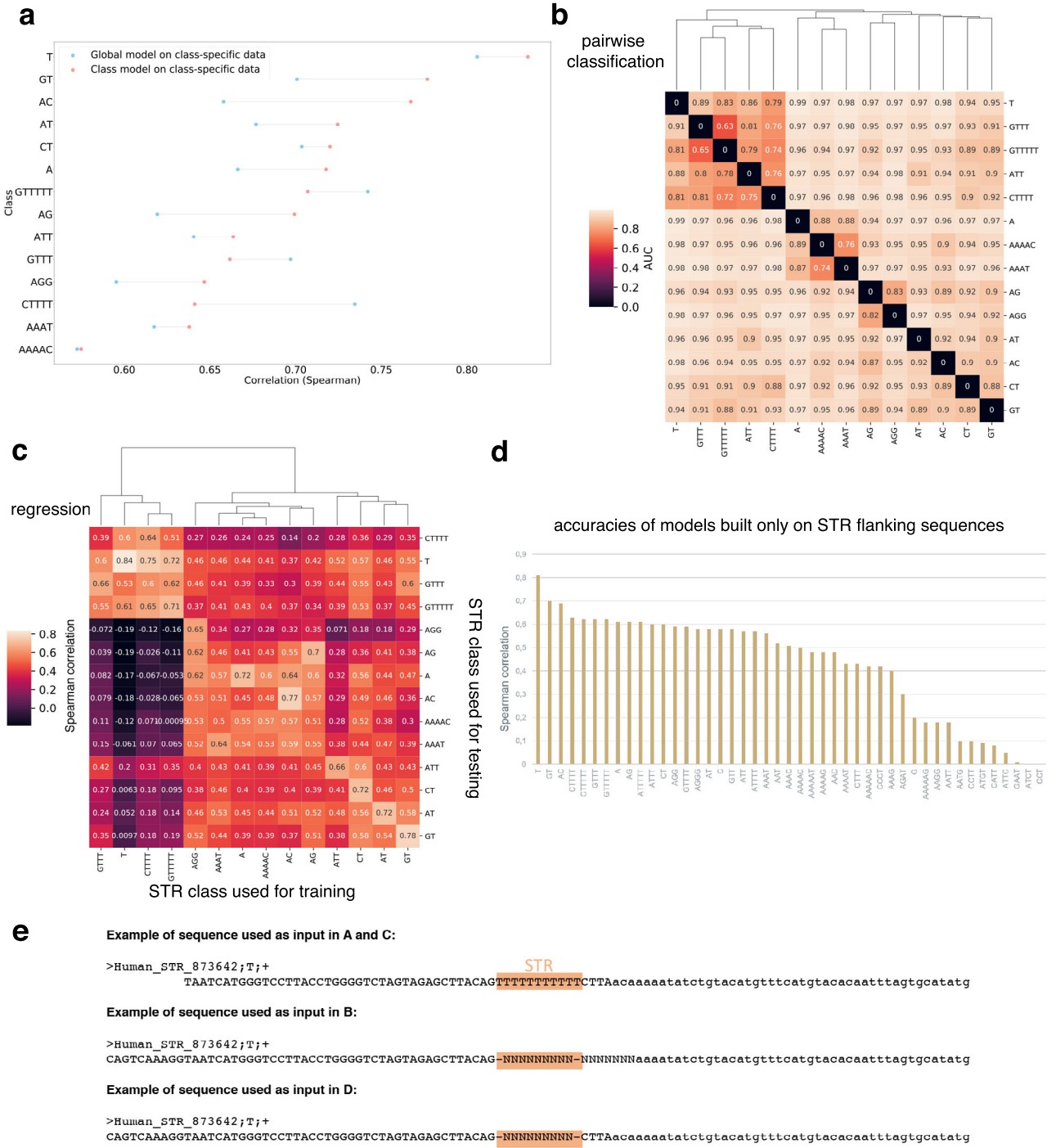

STR flanking sequences can classify STR classes, independently of the DNA repeated motif. It was previously shown that 50-bp-long sequences flanking $(AC)_n$ have evolved unusually to create specific nucleotide patterns[45]. To determine if such specific patterns hold true for other STRs, we sought to classify STRs based only on their 50 bp surrounding sequences. We trained a CNN model to classify pairs of STR classes (Supplementary Fig. 7). To avoid any problem due to the imprecise definition of STR boundaries, we masked the seven bases located downstream the STR 3′ ends (see "Methods"). In that case, model performance is evaluated by the Area Under the ROC (Receiver Operating

Characteristics) curve (AUC, Fig. 5b). The AUCs obtained in these pairwise classifications were very high (AUC > 0.7, Fig. 5b), with the notable exceptions of $(GTTT)_n$ vs. $(GTTTTT)_n$ (see below). Thus, STRs can be accurately distinguished, one from each other, using only 50-bp flanking sequences, and not the DNA repeated motif, even in the case of complementary STRs, such as $(AC)_n$ and $(GT)_n$ (Fig. 5b).

**Deep learning models unveil the key role of STR flanking sequences**. To further probe the sequence-level features for transcription initiation at STRs, we decided to build a model for

**Fig. 5 Probing STR sequences with CNN models. a** Comparison of the accuracies of global vs. class-specific models to predict transcription initiation levels at STRs. A model was learned on all STR sequences, irrespective of their class, and tested on each indicated STR class (accuracies obtained in each case, as Spearman $\rho$, is shown as blue points). Distinct models were also learned for each indicated class, without considering others (accuracies are shown in red). In total, 14 STR classes are shown as representative examples. Example sequence used as input is shown in E. **b** CNN-based pairwise classification of STRs using only STR flanking sequences (see "Methods" section). The pairs are defined by the line and the column of the matrix (e.g., the bottom left tile represents a classification task between T flanking sequences and GT flanking sequences). The values displayed on the tiles correspond to AUCs measured on the test set with the model trained specifically for the task. Clustering was performed to group pairs of STRs according to AUCs. **c** CNN performances to predict transcription initiation levels at heterologous STRs evaluated as the Spearman correlation between predicted and observed CAGE signal. The heatmap represents the performance of one model learned on one STR class (rows) and tested either on the same or another class (columns). Clustering is also used to show which models are similar (high correlation) and which ones differ (low correlation). **d** CNN models were learned on flanking sequences. The models use as an input only the 50-bp-long sequences flanking the STR, with the DNA repeated motif being masked by 9Ns (vectors of zeros in the one-hot encoded matrix). **e** Example of sequence used as input for each analysis depicted in A, B, C, and D. The pink box highlights the STR. All STRs are replaced by 9Ns in B and D, no matter their lengths. Additional seven bases downstream STR 3' end are masked in B because this window can contain bases corresponding to the DNA repeat motif, a feature that can easily be learned for STR classification. See details in the "Methods" section.

each STR class with >5000 elements ($n = 47$). Here, CNN is again used in a regression task to predict the CAGE signal. Sequences spanning 50 bp around the 3' end of each STR were used as input. Longer sequences were tested without improving the accuracy of the model (Supplementary Fig. 8). These class-specific models achieved overall better performances than the global model tested on each STR class separately (Fig. 5a and Supplementary Fig. 9). The only exceptions were classes composed of repetitions of T ($(GTTTTT)_n$, $(GTTT)_n$, and $(CTTTT)_n$). In these cases, global and $(T)_n$-specific models achieved better performance than $(GTTTTT)_n$, $(GTTT)_n$, or $(CTTTT)_n$-specific models. These results have two explanations: (i) compared to $(T)_n$, these classes have less occurrences (18,707 for $(GTTTTT)_n$, 55,898 for $(GTTT)_n$ and 15,433 for $(CTTTT)_n$), making it hard to learn models for these classes and (ii) the classification AUCs to distinguish $(GTTTTT)_n$, $(GTTT)_n$ or $(CTTTT)_n$ from $(T)_n$ was among the lowest observed (Fig. 5b), suggesting the existence of common sequence features that can be used by global and $(T)_n$-specific models. Overall, we estimated that STR class-specific models were accurate for 14 STR classes ($\rho > 0.65$).

We anticipated that class-specific models should not be equivalent and could not be interchangeable. We formally tested this hypothesis by measuring the accuracy of a model learned on one STR class and tested on another one (Fig. 5c). We caution again the fact that the performance of an STR-specific model also depends on the number of sequences available for learning. As observed earlier, the best accuracy is obtained with $(T)_n$, which are overrepresented in our catalog. Overall, the performance of one model tested on another STR class drastically decreases (Fig. 5c), revealing the existence of STR class-specific features predictive of transcription initiation. We also noticed that several models achieved non-negligible performances on other STR classes (Spearman $\rho > 0.5$, Fig. 5c), implying that some features governing transcription initiation at STRs are conserved between these STR classes. Thus, CNN models identified both common and specific features able to predict transcription initiation at STRs.

Our results unveil the importance of STR flanking sequences. We then evaluated the contribution of the sole surrounding sequences in transcription initiation prediction and built a model considering only these sequences (50 bp upstream and downstream STR, masking the STR itself, Fig. 5e). These models were less accurate than the formers but accuracies were still high for several classes (Fig. 5d), confirming that surrounding sequences contain features for transcription initiation prediction. The observed decrease in accuracies (Fig. 5d) implies that the STR itself contains features, which are combined with others present in flanking regions to predict transcription initiation. Remember that the CAGE signal predicted by our CNN models is normalized by

the length of the STR (see above), which makes them unable to assess the contribution of STR length in transcription initiation.

**Several sequence-level features predicting transcription initiation at STRs are conserved between human and mouse.** To test whether transcription at STRs is biologically relevant, we relied on two criteria: conservation and association with diseases. First, we studied conservation in mouse.

The number of loci within each STR class differs in mouse and human HipSTR catalogs (Figs. 1b and 6a and Supplementary Fig. 10). We applied the strategy used in human to compute the CAGE signal (as mean raw tag count in STR ± 5 bp divided by STR length + 10 bp) in mouse using 397 CAGE libraries (Fig. 6b). As observed in human, several STR classes were associated with CAGE signal. This signal appears lower than in human (compare Figs. 1d and 6b). This might be due to the fact that mouse CAGE data are small-scaled in terms of the number of reads mapped and diversity in CAGE libraries, compared to human CAGE data[2], making the mouse CAGE signal at STRs probably less accurate than the human one.

We nonetheless tested the correlation of the human and mouse CAGE signals at orthologous STRs. Orthologous STRs were identified converting the mouse STR coordinates into human coordinates with the UCSC liftover tool (see "Methods"). We intersected the coordinates of human STRs with that of orthologous mouse STRs and computed the Pearson correlation between the CAGE signal observed in human and that observed in mouse on the same strand ($n = 18,072$). In that case, Pearson's $r$ reaches ~0.87 (Spearman $\rho \sim 0.51$), suggesting that transcription at STRs is indeed conserved between mouse and human. As expected, no correlation was observed ($r < 0.01$) when randomly shuffling one of the two vectors or when correlating the signals of 18,072 randomly chosen mouse and human STRs.

We then built a CNN model to predict the CAGE signal at mouse STR classes corresponding to the 14 classes shown in Fig. 5a (Fig. 6c, green dots). The performances of the models ranged from ~0.4 to ~0.8, demonstrating that, as observed for human STRs, transcription at several mouse STR classes can be predicted by sequence-level features. A notable exception is $(CTTTT)_n$ with Spearman $\rho < 0.2$ (see below). The mouse models were overall less accurate than human models (Fig. 6c, compare red and green dots), likely due to differences in the quality of the CAGE signal (i.e., predicted variable), as mentioned above.

We then tested whether the sequence features able to predict STR transcription initiation were conserved between mouse and human. We specifically tested the performances of models learned in one species and tested on another one (Fig. 6c, blue

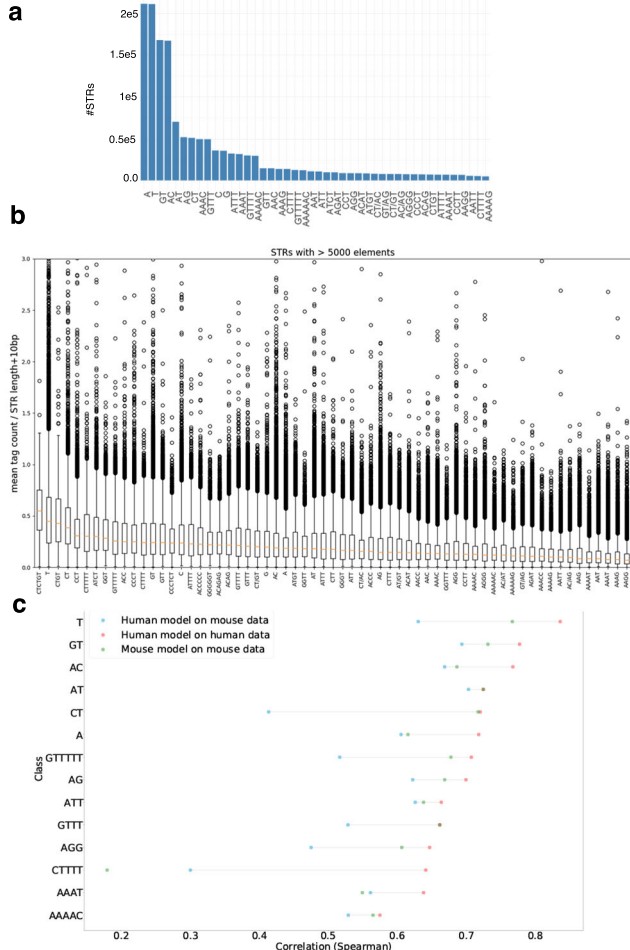

**Fig. 6 STR transcription initiation in mouse. a** Number of mouse STRs per class. For sake of clarity, only STR classes with >5000 loci are shown. **b** CAGE signal at mouse STR classes with >5000 loci. CAGE signal was computed as in Fig. 1d. Boxplots are defined as in Fig. 1d. **c** Testing the accuracy of CNN models built in human and tested in mouse for different STR classes. Performances of the models are assessed by computing the Spearman $\rho$ between (i) CAGE signal observed in mouse and signal predicted by a model learned in human (blue dots), (ii) CAGE signal observed in mouse and signal predicted by a model learned in mouse (green dots), and (iii) CAGE signal observed in human and signal predicted by a model learned in human (red dots).

dots and Supplementary Fig. 11). For all STR classes tested, the Spearman correlation between the signal predicted by the human model and the observed mouse signal was >0.4 (Fig. 6c), implying that several features are conserved between human and mouse. For some classes (e.g., $(A)_n$, $(AC)_n$, $(AAAT)_n$), the human and mouse models even appeared equally efficient in predicting transcription initiation in mouse (Fig. 6c, green and blue dots are close), indicative of strong conservation of predictive features. For other classes (e.g., $(CT)_n$, $(AGG)_n$), the performance of the human model was lower than that obtained with the mouse model when tested on mouse data (Fig. 6c, green and blue dots are distant). Thus, specific features also exist in mouse that were not learned in human sequences. Likewise, human-specific features also exist (Supplementary Fig. 11). In the case of $(CTTTT)_n$, the human model performs better than the mouse one (Fig. 6c). This effect is likely due to the number of examples, which is higher in human ($n = 15,433$) than in mouse ($n = 10,494$). Overall, we conclude

that several features predictive of transcription initiation at STRs are conserved between human and mouse and that the level of conservation also varies depending on STR classes.

**ClinVar pathogenic variants are found at STRs with high transcription initiation level.** Second, we evaluated the potential implication of transcription initiation at STRs in human diseases and used the ClinVar database, which lists medically important variants[46]. We found that STRs harboring ClinVar variants, located in a window encompassing STR ± 50 bp ($n = 34,578$), are associated with high CAGE signal compared to STRs without variants ($n = 3,068,280$, Fig. 7a), indicative of potential biological and clinical relevance for transcription initiation at STRs. Looking at the clinical significance of the variants, as defined in the ClinVar database, we indeed noticed that STRs associated with pathogenic variants exhibit stronger transcription initiation than STRs associated with other variants (Fig. 7b and Supplementary Fig. 12). STRs could be associated with more or less variants linked to a given disease than expected by chance (adjusted $P$ value < 5e-3, Supplementary Data 2) but no clear association with a specific clinical trait was noticed.

We initially sought to identify representations of sequence motifs captured by CNN first layer filters using a strategy inspired by Maslova et al.[47] and identified several influential first layers correlating with JASPAR PMW scores (see "Methods" section and Supplementary Tables provided here at https://gite.lirmm.fr/ibc/deepSTR//first_layer_interpretation). However, it is important to remember that our models were optimized to predict CAGE signal, not to learn interpretable representations from input DNA sequences. Koo and Eddy have indeed demonstrated that tackling these two questions—prediction and interpretation —requires distinct CNN architectures, in particular adapting max-pooling and convolutional filter size[48]. At present, our models likely learn partial motifs and do not limit the ability to learn full interpretable motifs in deeper layers. We then used a perturbation-based approach[49] and randomly created in silico mutations to identify key positions of the models (see "Methods" section). Random variations were directly introduced into STR sequences, and predictions were made on these mutated sequences using the CNN model-specific of the STR class considered. The impact of the variation was then assessed as the difference between the predictions obtained with mutated and reference sequences. Same analyses were performed with ClinVar variants (Fig. 7c and Supplementary Fig. 13). Key positions were defined as positions, which, when mutated, have a strong impact on the prediction changes (i.e., high variance), being either positive or negative. As shown in Fig. 7c, for both random and ClinVar variants, the most important positions appeared located around STR 3' end (−15 bp/+30 bp) and their distribution is skewed toward the sense orientation of the transcripts. Strikingly, a significant proportion of ClinVar variants are located in the immediate vicinity of the STR 3' end (Fig. 7d). Hence, the most important positions identified by our models correspond to positions with high occurrences of ClinVar variants (Fig. 7c, d). However, neither the distribution nor the impact of variants appears linked to their pathogenicity because similar results are observed for both benign and pathogenic variants (Supplementary Fig. 14). Note that ClinVar variants are also concentrated around assigned CAGE peak summits and all identified CAGE peak summits (Supplementary Fig. 15). Overall, we conclude that the pathogenicity of ClinVar variants appears to be linked to the transcription initiation level at the targeted STR rather than to the position of the variation or its impact on prediction.

Finally, as machine-learning approaches only unveil correlation between predictive and predicted features, not direct causation, we

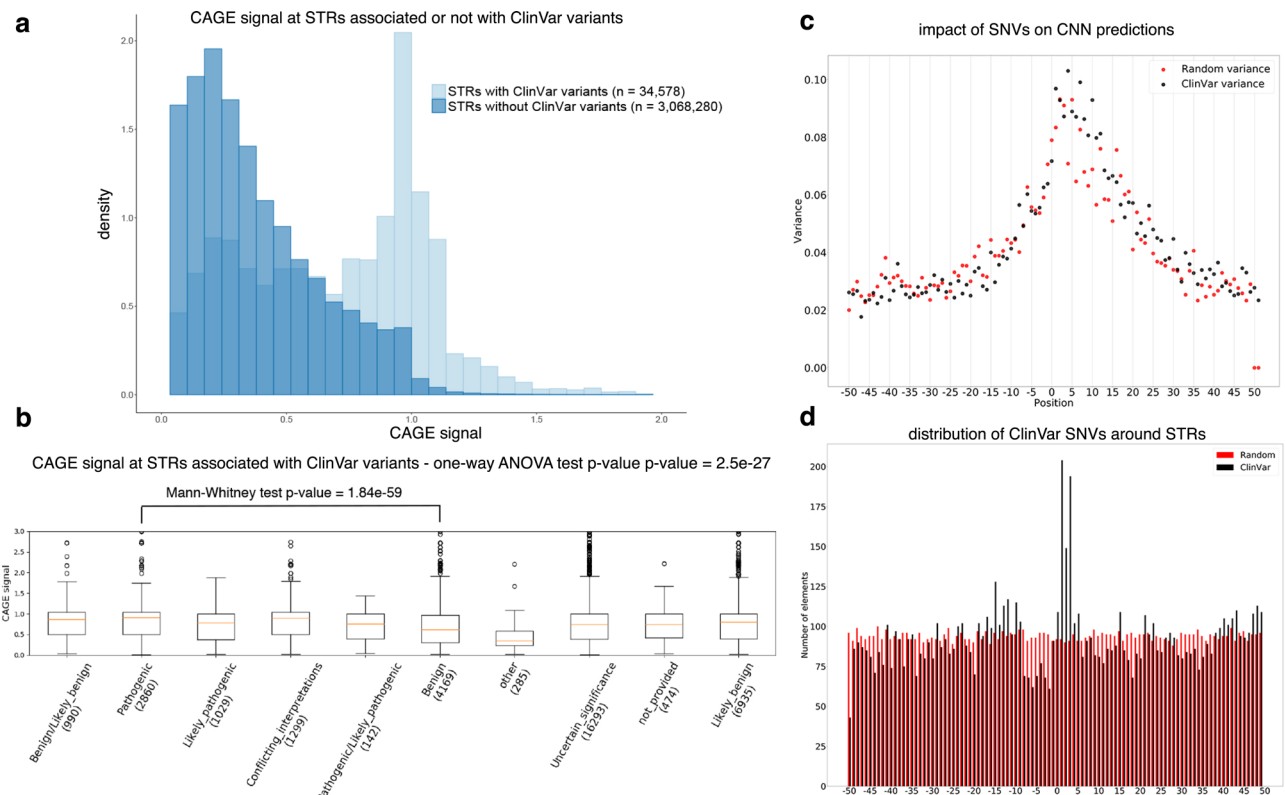

**Fig. 7 ClinVar variants at STRs. a** CAGE signal distribution of STRs associated (light blue) or not (dark blue) with at least one ClinVar variant. The number of STRs considered in each case is indicated in the bracket. **b** CAGE signal ($y$ axis) at STRs associated with ClinVar variants ordered according to their clinical significance ($x$ axis). The number of variants considered for each ClinVar class is indicated in the bracket. A one-way ANOVA test was used to assess overall statistical differences ($P$ value = 2.5e-27). Pairwise comparisons using one-sided Mann–Whitney rank tests were also performed ($P$ values are indicated in Supplementary Fig. 12). Boxplots are defined as in Fig. 1d. **c** Impact of the changes induced by ClinVar (black) and random (red) variants on CNN predictions. Predictions are made on the hg19 reference sequence and on a mutated sequence, containing the genetic variants. Changes are then computed as the difference between these two predictions (reference - mutated, Supplementary Fig. 13) and their impact is measured as their variance at each position around STR 3' end ($x$ axis). To keep sequences aligned, only single nucleotide variants (SNVs) were considered. **d** Distribution of ClinVar (black) and random (red) variants around STR 3' end. The number of variants and their position relative to STR 3' end (position 0) are indicated on the $y$ axis and $x$ axis, respectively. A Kolmogorov–Smirnov test was used to assess statistical significance between the distribution of ClinVar variants and that of random variations ($P$ value = 2.95e-11).

sought to determine whether the features learned by our models correspond to sequence-level instructions for transcription initiation. We looked for gene TSSs located at STRs and harboring variants acting as eQTLs for the corresponding genes, in a scenario similar to that described by Bertuzzi et al. in the case of a minisatellite and the NPRL3 gene[20]. Gene expression is considered here as a proxy for the measure of transcription initiation at STRs. In that scenario, if our models capture instructions for expression, the difference of the predictions made by our models for the reference and the alternative alleles should have the same sign as the eQTL slope (i.e., gene expression increase (slope > 0) or decrease (slope < 0)) more often than expected by chance. First, to identify STRs potentially acting as TSSs, we selected STRs located in gene promoters (considering 1 kb around FANTOM CAT gene start). We only considered models with accuracy >0.7 (Fig. 5c). Second, based on our results depicted in Fig. 7c, we selected GTEx eQTLs located in a −15-bp/+30-bp window around STR 3' end and linked to the expression of the genes associated with STRs in the first step. These selections yielded 86 cases of STR sequence variations linked to gene expression by eQTL. Of note, we first thought to use FANTOM CAT transcript TSSs directly, instead of gene TSSs, but only one case was identified with prediction error (measured as the absolute value of the difference between the predicted and the observed CAGE signals) < 0.2. The alternative alleles corresponding

to the selected eQTLs were inserted into their cognate STR sequences and a prediction was made for this modified sequence. The sign of the difference between the two predictions (alternative - reference) was compared to the sign of the eQTL slope. We counted the number of times these signs were identical or different (Supplementary Fig. 16). The prediction errors of the models for these 86 STRs were also computed in the case of the reference genome (Supplementary Fig. 16). As shown in Supplementary Fig. 17, when predictions are accurate on the reference genome (error ≤ 0.2), the models are able to predict the impact of variants on expression i.e., in most cases, the sign of the difference between the predictions made with the alternative and predictive alleles is similar to that of the eQTL slope. Importantly, this is no longer observed when the models poorly perform (error > 0.2). Binomial tests were used to statistically assess the relevance of these findings. Thus, when accurate, our models are able to predict the effects of eQTLs, supporting a causal relationship between the predictive and the predicted variables rather than a mere correlation.

## Discussion

We report here the discovery of widespread transcription initiation at STRs in human and mouse. These results extend previous findings[30–33] and reveal that, in addition to being the passenger of

host RNAs initiating at their own TSSs[30–33], STRs can also initiate the transcription of distinct and autonomous RNAs. The next main issue is to determine the role(s) of these transcripts. RNA species can be functionally categorized according to transcriptional directionality[39]. In the case of STRs, transcription directionality appears to depend on the STR class (Fig. 4b). It is thus likely that RNAs initiating at STRs fulfill distinct functions and many hypotheses could be proposed at this stage. For instance, 10,727 CAGE peaks mapped at STRs correspond to TSSs of FANTOM CAT transcripts (Supplementary Data 1), extending the findings made by Bertuzzi et al. in the case of a minisatellite and the NPRL3 gene[20] to STRs. Many RNAs initiating at STRs may also correspond to noncoding RNAs, as for instance enhancer RNAs (Supplementary Data 1). As could have been anticipated given the distinction of enhancers and promoters based on CpG dinucleotide[50], FANTOM CAT transcripts mostly initiate at GC-rich STRs, while enhancer RNAs more often correspond to A/T-rich STRs (Supplementary Data 1). Another possible function is provided by $(T)_n$, which are over-represented in eukaryotic genomes[51] and have been shown to act as promoter elements by depleting repressive nucleosomes[52]. As a consequence, $(T)_n$ can increase transcription of reporter genes in similar levels to TF-binding sites[53]. The findings that $(A)_n$ and $(T)_n$ represent distinct directional signals for nucleosome removal[54] are very well compatible with differences observed in flanking sequences (Fig. 5b) and directional transcription (Fig. 4b), both able to create asymmetry at $(A)_n$ and $(T)_n$. Besides, we show that most CAGE tags initiating at STRs remain nuclear (Fig. 4a). This observation suggests that, similar to other repeat-initiating RNAs[55,56], RNAs initiating at STRs could also play roles at the nuclear/chromatin levels, for instance in DNA topology[56,57]. Note that we also calculated the enrichment of STR classes in FANTOM CAT biotypes (Supplementary Data 3). The strongest enrichments correspond to $(A)_n$, $(AT)_n$, and $(AAAT)_n$ at enhancers, which are known to be GC-poor sequences compared to promoters for instance[50]. It also remains to clarify whether STR-associated RNAs or the act of transcription per se is functionally important[10]. Dedicated experiments are now required to formally identify the biological functions linked to the transcription of each STR class. These experiments are all the more warranted as STR transcription is associated with clinically relevant genomic variations (Fig. 7).

One key finding of our study is the discovery that STR flanking sequences are not inert but rather contain important features that play critical roles in their biology, as previously suspected[45]. These results call for the development of novel methods able to take these sequences into account in order to revisit STR mapping/genotyping and integrate SNVs located in STR vicinity. These methods should have broad applications in various fields of research and medicine, from forensic medicine to population genetics for instance. STR length variations have notably been shown to influence gene expression and, similar to eQTLs, several eSTRs have been identified[58,59]. Their exact mode of action still remains largely elusive but, the majority of eSTRs appear to act by global mechanisms, in a tissue-agnostic manner[58]. Interestingly, some eSTRs have strand-specific effects[58], which is again compatible with the possible sources of asymmetry unveiled by our study (i.e., flanking sequences and directional transcription). Using transcription initiation level at STRs, as predicted by our CNN models for instance, coupled with length variations[58,59], may help to take into account the impact of genetic variants located in sequences surrounding STRs[60], and to refine eSTR computations. Results depicted in Supplementary Figs. S16 and S17 show that CNN models can indeed refine eSTR computations by simply re-assigning eQTLs as eSTRs.

There are still several ways to improve our CNN models. Notably, to avoid any bias linked to the CAGE noise signal observed along STRs, we decided to predict a signal normalized by the STR length. Therefore, our models do not allow to properly assess the contribution of STR length in transcription, although it clearly represents the most studied feature of STRs[21,58,59]. Note that simply increasing the quality of the reads considered (using Q20 instead of Q3 filter) yields sparse data and decreases the performance of our model. A new computation of the CAGE signal aimed at removing "noise" at STRs could be developed. This may also help develop tissue-specific CNN models, which will only use CAGE data[44]. Besides, the same architecture was used for all STR classes while achieving different accuracies (Fig. 5a, c). These results cannot be merely explained by the number of STR sequences available for training because swapping the models for training and testing demonstrated the existence of STR class-specific features predictive of transcription initiation (Fig. 5c). It is rather possible that the chosen architecture may not be optimal for all STRs, as illustrated by the design of a global model with overall good performance, but very distinct accuracies depending on the STR class (Fig. 5a). Our CNN architecture was initially optimized on the $(T)_n$ class, which represents the most abundant class ($n = 766,747$). Because each STR class harbors sequence specificities including in flanking sequences, hyperparameters, such as convolutional filter sizes, their number, and/or max-pooling, could be adapted to each STR class. These hyperparameters have indeed already been shown to influence the results of CNN models as well as their interpretation[48].

More broadly, the same rationale could be applied to other methods aimed at predicting CAGE signal along the genome[44], distinguishing biological entities (genes, enhancers, …), genomic segments[61,62], and/or isochores[63] based on their sequence features. Building a general model increases the risk of designing a model suited for the most represented elements, not for the others. Notably, promoters and enhancers can be distinguished by different CpG content, the presence of polyA signal and of 5' splice sites[40,50], as well as different transcription factor combinations[3,64]. It is therefore likely that the same filters will not apply similarly to predict transcription in both cases and that one may want to develop a specific model for each of these entities to increase the accuracy of the predictions.

The prediction of transcription initiation based solely on sequence features has long been studied, especially using CAGE data[65,66]. The high accuracy achieved by CNN models for this task, as illustrated in this study or in refs. [43,44,47], as well as the development of methods aimed at interpreting this type of statistical models[48,49,67,68], will certainly accelerate the achievement of this goal, which becomes more than ever "a realistic short-term objective rather than a distant aspiration"[66].

## Methods

**Data and bioinformatic analyses.** The bedtools window[69] was used to look for CAGE peaks (coordinates available at http://fantom.gsc.riken.jp/5/datafiles/phase1.3/extra/CAGE_peaks/hg19.cage_peak_coord_permissive.bed.gz) at STRs ± 5bp (catalog available at https://github.com/HipSTR-Tool/HipSTR-references/raw/master/human/hg19.hipstr_reference.bed.gz) as follows:

```
windowBed -w 5 -a hg19.hipstr_reference.bed
-b hg19.cage_peak_coord_permissive.bed
```

As a comparison, random intervals were generated using bedtools shuffle[69].

```
shuffleBed -i hg19.hipstr_reference.bed -g
hg19.chrom.sizes -excl hg19.hipstr_reference.
bed -seed 927442958 > hg19.hipstr_reference.
shuffled.bed
```

Similar analyses were performed using mouse STR catalog (available at https://github.com/HipSTR-Tool/HipSTR-references/blob/master/mouse/mm10.hipstr_reference.bed.gz) liftovered to mm9 using UCSC liftover tool[70]:

```
liftover mm10.hipstr_reference.bed
mm10ToMm9.over.chain.gz mm9.hipstr_
reference.bed unlifted.bed
```

To compute the CAGE signal, we used raw tag count along the genome with a 1-bp binning and Q3 quality mapping filter. At each position of the genome, the mean tag count across 988 libraries for human and 387 for mouse was computed. The values obtained at each position of a window encompassing the STR ± 5 bp were then summed and normalized (i.e., divided by the STR length + 10 bp) to limit the impact of the CAGE noise signal observed along STRs. CAGE signals at human and mouse STRs are available at https://gite.lirmm.fr/ibc/deepSTR, as, respectively, hg19.hipstr_reference.cage.bed and mm9.hipstr_reference.cage.bed (The CAGE signal is indicated in the 5th column). The fasta files (500 bp around STR 3' end) to build our models are also available at the same location as hg19.hipstr_reference.cage.500bp.around3end.fa and mm9.hipstr_reference.cage.500bp.around3end.fa. CNN models use as input 101-bp-long sequences centered around STR 3' ends.

The bedtools intersect[69] was used to distinguish intra- and intergenic STRs, intersecting their coordinates with that of the FANTOM gene annotation (available at https://fantom.gsc.riken.jp/5/suppl/Hon_et_al_2016/data/assembly/lv3_robust/FANTOM_CAT.lv3_robust.bed.gz).

Coordinates of FANTOM CAT robust transcripts and FANTOM enhancers can be found, respectively, at these URLs: transcripts [http://fantom.gsc.riken.jp/5/suppl/Hon_et_al_2016/data/assembly/lv3_robust/FANTOM_CAT.lv3_robust.gtf.gz] and enhancers [https://fantom.gsc.riken.jp/5/datafiles/latest/extra/Enhancers/human_permissive_enhancers_phase_1_and_2.bed.gz]. ENCODE RNAPII ChIP-seq bed files can be downloaded following these links: GM12878, H1-hESC [http://hgdownload.cse.ucsc.edu/goldenpath/hg19/encodeDCC/wgEncodeAwgTfbsHaibH1hescPol2V0416102UniPk.narrowPeak.gz], HeLa-S3 [http://hgdownload.cse.ucsc.edu/goldenpath/hg19/encodeDCC/wgEncodeAwgTfbsHaibHelas3Pol2Pcr1xUniPk.narrowPeak.gz] and K562.

Expression data used to determine the nucleo-cytoplasmic distribution of CAGE peaks can be found at http://fantom.gsc.riken.jp/5/datafiles/latest/extra/CAGE_peaks/hg19.cage_peak_phase1and2combined_tpm_ann.osc.txt.gz.

Orthologous STRs were identified using UCSC liftover tool[70] and the mm9ToHg19.over.chain.gz file.

For eQTLs, we used GTEx V7 data [https://storage.googleapis.com/gtex_analysis_v7/single_tissue_eqtl_data/GTEx_Analysis_v7_eQTL.tar.gz].

All statistical tests were performed with R (*wilcoxon.test*, *fisher.test*) or Python (*scipy.stats.f_oneway*, *scipy.stats.mannwhitneyu*, *scipy.stats.kstest*), as indicated. When indicated, P values were corrected for multiple testing using R *p.adjust* (method="fdr").

**Evaluating mismatched G bias at Illumina 5' end CAGE reads**. Comparison between Heliscope vs. Illumina CAGE sequencing was performed as in de Rie et al.[38]. Briefly, ENCODE CAGE data were downloaded as bam files (using the following url [http://hgdownload.cse.ucsc.edu/goldenpath/hg19/encodeDCC/wgEncodeRikenCage/] ('*NucleusPap*' files) and converted into bed files using samtools view[71] and UNIX awk:

```
samtools view file.bam |awk '{FS="\t"}BEGIN{OFS="\t"}{if
($2=="0") print $3,$4-1,$4,$10,$13,"+"; else if($2=="16") print
$3,$4-1,$4,$10,$13,"-"}' >file.bed
```

The bedtools intersect[69] was further used to identify all CAGE tags mapping a given position. The UNIX awk command was used to count the number and type of mismatches:

```
intersectBed -a positions_of_interest.bed -b file.bed -wa
-wb -s |awk '{if(substr($11,1,6)=="MD:Z:0" && $6=="+") print
substr($10,1,1)}'| grep -c "N"
```

with N = {A, C, G or T}, positions_of_interest.bed being coordinates of CAGE peaks assigned to genes, or that located at pre-miRNA 3' ends, or peaks associated with STRs. The file.bed corresponds to the Illumina CAGE tag coordinates.

The absence of mismatch focusing on the plus strand was counted as:

```
intersectBed -a positions_of_interest.bed -b file.bed -wa
-wb -s |awk '{if(substr($11,1,6) !="MD:Z:0" && $6=="+") print
$0}'|Êwc -l
```

As a control, we used the 3' end of the pre-miRNAs, which were defined, as in de Rie et al.[38], as the 3' nucleotide of the mature miRNA on the 3' arm of the pre-miRNA (miRBase V21 [ftp://mirbase.org/pub/mirbase/21/genomes/hsa.gff3]), the expected Drosha cleavage site being immediately downstream of this nucleotide (pre-miR end + 1 base).

**Cap-Trapping MinION sequencing**. A549 cells were grown in Dulbeccoõs modified Eagle medium (DMEM) supplemented with 10% fetal bovine serum (FBS). A549 cells were washed with PBS. The RNAs were isolated by using RNeasy kit (QIAGEN). The poly-A tail addition to A549 total RNA was carried out by poly-A polymerase (PAPed RNA). The cDNA synthesis was carried out by using 5 μg of total RNA or 1 μg of PAPed RNA with RT primer (5-TTTTTTTTTUUUTTTTTVN-3) by PrimeScript II Reverse Transcriptase (TaKaRa Bio). The full-length cDNAs were selected by the Cap Trapper method[72]. After the ligation of 5' linker, cDNAs were treated with USER enzyme to shorten the poly-T derived from RT primer. After SAP treatment, a 3' linker was ligated to the cDNAs. The linkers used in the library preparation were prepared as in ref. [72] with oligos provided in Supplementary Table 1. As for the 3' linker, after annealing step, the UMI complemental region (BBBBBBBB) was filled with Phusion High-Fidelity DNA polymerase (NEB) and dVTPs (dATP/dGTP/dCTP) instead of dNTPs. The second strand was synthesized using a second primer with KAPA HiFi HS mix (KAPA Biosystems). The double-stranded cDNAs were amplified using Illumina adapter-specific primers and LongAmp Taq DNA polymerase (NEB). After 16 cycles of PCR (8 min for elongation time), amplified cDNAs were purified with an equal volume of AMPure XP beads (Beckmann Coulter). Purified cDNAs were subjected to Nanopore sequencing library following manufacturerõs 1D ligation sequencing protocol (version NBE_9006_v103_revO_21Dec2016).

Nanopore libraries were sequenced by MinION Mk1b with R9.4 flowcell. Sequence data were generated by MinKNOW 1.7.14. Basecalling was processed by ÓAlbacore v2.1.0 basecaller software provided by Oxford Nanopore Technologies to generate fastq files from FAST5 files. To prepare clean reads from fastq files, adapter sequence was trimmed by Porechop v0.2.3. Data were deposited on DNA Data Bank of Japan Sequencing Read Archive (accession number: DRA010491). The mapping computational pipeline used a prototype of primer-chop available at https://gitlab.com/mcfrith/primer-chop. The precise methods and command lines are provided as Supplementary Methods. Data were first mapped on hg38 reference genome and liftovered to hg19 for analyses.

**Directionality score**. We collected CAGE signal at each STR of the HipSTR catalog (see above). When a signal was detected on both (+) and (−) strands, we computed the directionality score for each STR using the following formula:

$$\frac{(CAGE\ signal\ on\ the\ (+)\ strand\ -\ CAGE\ signal\ on\ the\ (-)\ strand)}{(CAGE\ signal\ on\ the\ (+)\ strand\ +\ CAGE\ signal\ on\ the\ (-)\ strand)}$$

The CAGE signal was computed as explained above. A score equals to 1 or −1 indicates that transcription is strictly oriented towards the (+) or (−) strand, respectively. A score close to 0 indicates that the transcription is balanced and that it occurs equally on the (+) and (−) strands.

U1 PWM was built using MEME[73] and sequences encompassing −3/+10 bp around FANTOM CAT 5' donor splice sites (exon 3' end). We then used this PWM and FIMO[74] to scan 2kb regions centered around 3' ends $(T)_n$ STRs (considering the top 50,000 sequences with the highest CAGE signal) and FANTOM CAT TSSs. For polyA sites, we used the UCSC track corresponding to the predictions made by Cheng et al.[75], as a bed file and used it in bedtools intersect[69] to look at polyA site distribution in regions encompassing 1 kb around $(T)_n$ 3' ends (top 50,000 with the highest CAGE signal) and FANTOM CAT TSSs.

**Convolutional neural network**. CNN architecture is described in Supplementary Fig. 7. To build a CNN, we needed aligned sequences of equal length. However, as shown in Supplementary Fig. S1, CAGE peaks are scattered along STRs. We thus decided to align the sequences on STR 3' ends, as defined by the CAGE data. HipSTR indeed provides a catalog built on the (+) strand but CAGE data are stranded data (see Fig. 1a). CAGE thus allows to orientate each STR of the HipSTR catalog as exemplified here:

**HipSTR catalog (see hg19.hipstr_reference.bed):**
chr1 10001 10468 6 78 Human_STR_1 AACCCT
**Same STR with CAGE data (see hg19.hipstr_reference.cage.bed made available at https://gite.lirmm.fr/ibc/deepSTR)**
chr1 10001 10468 Human_STR_1; AACCCT; + 0.410901 +
chr1 10001 10468 Human_STR_1; AACCCT; − 0.354298 −

It is then possible to determine the 3' end of each STR according to the strand considered (here 10468 on the (+) strand and 10002 on the (−) strand). This procedure almost doubles the number of elements in each class.

Sequences spanning 50 bp around the 3' end of each STR were used as input unless otherwise stated (see Fig. 5e). Longer sequences were tested without improving the accuracy of the model (Supplementary Fig. 8). Note that only 89,189 STRs (out of 1,620,030, ~5.5%) are longer than 50 bp and, only in these few cases, the sequence located upstream STR 3' end only corresponds to the STR itself. The parameters of the model were determined by brute force algorithms using a grid search approach. This approach makes a complete search over all hyperparameters (number of layers, number of neurons, activation functions, different learning rates, shape of convolutional kernels, number of convolutional filters, …). The grid search algorithm trains and tests all possible models with all combinations of parameters and returns the most accurate model. The model was implemented in PyTorch. The source code of the model, alongside scripts and Jupyter notebooks are available at https://gite.lirmm.fr/ibc/deepSTR.

In order to minimize overfitting, droupout is added to the fully connected layers (probability of droupout = 0.30). The training pipeline is described in Supplementary Fig. 7: we separate training, testing, and validation datasets prior to model training, and these sets are stored on disk. This allows us to carry out analyses on held-out data that has never been seen by the models. We stop the training once the loss function calculated on the validation set drops for five consecutive epochs (early stopping). Relatively good performances on mouse datasets (Fig. 6c) show that the model generalizes well to unknown CAGE data. Our models were optimized to predict CAGE signal and cannot, as such, be applied to other types of data. However, the methodology used here is generic and could be

applied to other types of data as long as one can associate a numeric signal to a specific genomic region.

To make sure that our models do not overfit due for instance to homologous sequences present in both train and test sets, we used BLASTn[76] to look for homology between $(T)_n$ sequences of the test and train sets. The model learned on $(T)_n$ STRs was used because it is the most accurate and therefore the more likely to overfit. We found 102,209 sequences from the test set with >60% query cover and >80% identity with at least one sequence of the train set. We separated these sequences (test set #1, homologous sequences) from the rest of the test set (test set #2, 121,808 nonhomologous sequences). We then computed Spearman correlations between the predicted and the observed CAGE signals using these two test sets: 0.73 with test set #1 and 0.78 with test set #2. In both cases, correlations decreased, as compared to correlation computed with the whole test set (0.84). This decrease is due to differences in CAGE signal distribution between the whole test set, test set #1 and #2 (Supplementary Fig. 18) likely linked to mapping issues. However, model performance measured on test set #2 was greater than that obtained with test set #1. This is in contrast to what is expected in the case of model overfitting due to sequence homology. We then concluded that homology observed between train and test sets is not sufficient to make the model overfit.

For comparison to the baseline model, we computed the correlation between the observed CAGE signal and randomized CAGE signal (equivalent to a predictor that returns a random value drawn from observed values). Randomization was repeated ten times and Spearman correlation was invariably close to 0 (absolute value ($\rho$) < 5e-4).

The models are provided at https://gite.lirmm.fr/ibc/deepSTR. They can be used to predict transcription initiation level at STRs using a fasta file. Likewise, impact of genetic variations can be assessed by comparing the predictions obtained for instance with reference and mutated sequences (see Fig. 7 and Supplementary Fig. 17).

**Classification**. The CNN model can also be set up for a classification task (Fig. 5b and Supplementary Fig. 7). In that case, the only difference with the regression model is the last neuron in the last fully connected layer. The classifier CNN uses the same training method. The data are also prepared by separate scripts before training is done and stored on disk. All analyses resulting from the classification are performed on the test sets to avoid optimistic bias in accuracy estimation. Note that 7 bp downstream STR 3' end were masked and replaced by Ns (Fig. 5e) because we noticed that this window can contain bases corresponding to the DNA repeat motif, a feature that can easily be learned by a CNN. The sequences used as input, for classification using flanking sequences only (Fig. 5d), are centered around STR 3' end and consist of 50-bp-long upstream sequence + 9 Ns, which mask the STR itself +7 Ns + 43-bp-long downstream sequence (total length = 109 bp, Fig. 5e).

**Model swaps between human STR classes**. After models are trained on all STR classes, their weights are stored in a .pt file (following the PyTorch convention). Predictions were then computed on all test sets with all models.

**Model interpretation**. First, for each of the 14 models presented in Fig. 5, we measured the influence of each first layer filters by removing them iteratively and computing the accuracy of the model (Spearman correlation between observed and predicted CAGE signal) with the 49 remaining filters. We also computed an influence threshold by learning each CNN model ten times and computing a 95% confidence interval (CI). The threshold was calculated as log2(CI length/2). This allows to focus our analyses on key filters, with performance impact greater than what would have been obtained by chance, simply re-training the model. Influential first layer filters are then ranked according to their influence. Second, on the one hand, we used FIMO[74] to scan 101-bp-long sequences centered around STR 3' end (considering all STR sequences if $n < 10,000$ or 10,000 randomly chosen sequences otherwise) with JASPAR PWMs[77]. For each PWM, we identified a set of STR sequences harboring PWM hits. For each sequence, we kept the PWM maximal score found. On the other hand, we scanned the 10,000 STR sequences with influential first layer filters as defined in step #1 (using matrix multiplication as in convolution) and kept the maximal value obtained for each sequence. We then computed the correlation between JASPAR PWM scores and first layer filter scores. We reasoned that if a filter represents a partial PWM, their score should be correlated. The results of these analyses are provided as Supplementary Tables located on our git repository [https://gite.lirmm.fr/ibc/deepSTR//first_layer_interpretation].

**Predicting the impact of ClinVar variants**. ClinVar vcf file was downloaded January 8th 2019 from this url [ftp://ftp.ncbi.nlm.nih.gov/pub/clinvar/] and then converted into bed file. We looked for STRs associated with ClinVar variants (Fig. 7a) using bedtools window[69] as follows:

```
bedtools window -w 50 -a clinvar_mutation.bed -b str_coordinates.bed
```

Variants were directly introduced into STR sequences (± 50 bp) using Biopython[78] library and the *seq.tomutable()* function. To keep sequences aligned, we only considered single nucleotide variants (SNVs). CNN models were then used to predict the CAGE signal of the initial and mutated sequences. The change was

computed by the difference between the prediction obtained with the mutated sequence and that obtained with the reference sequence. To insert random variations (Fig. 7c, d), we created a mutation position map, which follows a uniform distribution (each position has an equal probability of receiving a mutation). Then, we took sequences in the database and mutated them one by one at a position taken from the mutation map. All possible mutations at the chosen position have an equal probability of occurrence (Fig. 7d).

**Reporting summary**. Further information on research design is available in the Nature Research Reporting Summary linked to this article.

## Data availability
The data that support this study are available from the corresponding author upon reasonable request. CAGE peaks coordinates [http://fantom.gsc.riken.jp/5/datafiles/phase1.3/extra/CAGE_peaks/hg19.cage_peak_coord_permissive.bed.gz]; human STR catalog [https://github.com/HipSTR-Tool/HipSTR-references/raw/master/human/hg19.hipstr_reference.bed.gz]; mouse STR catalog [https://github.com/HipSTR-Tool/HipSTR-references/blob/master/mouse/mm10.hipstr_reference.bed.gz]; CAGE signals at human and mouse STRs, alongside fasta sequence files, are available on our git repository [https://gite.lirmm.fr/ibc/deepSTR]; FANTOM gene annotation [https://fantom.gsc.riken.jp/5/suppl/Hon_et_al_2016/data/assembly/lv3_robust/FANTOM_CAT.lv3_robust.bed.gz]; Coordinates of FANTOM CAT robust transcripts [http://fantom.gsc.riken.jp/5/suppl/Hon_et_al_2016/data/assembly/lv3_robust/FANTOM_CAT.lv3_robust.gtf.gz] and FANTOM enhancers [https://fantom.gsc.riken.jp/5/datafiles/latest/extra/Enhancers/human_permissive_enhancers_phase_1_and_2.bed.gz]; ENCODE RNAPII ChIP-seq bed files: GM12878 [http://hgdownload.cse.ucsc.edu/goldenpath/hg19/encodeDCC/wgEncodeAwgTfbsUniform/wgEncodeAwgTfbsHaibGm12878Pol2Pcr2xUniPk.narrowPeak.gz], H1-hESC [http://hgdownload.cse.ucsc.edu/goldenpath/hg19/encodeDCC/wgEncodeAwgTfbsHaibH1hescPol2V0416102UniPk.narrowPeak.gz], HeLa-S3 [http://hgdownload.cse.ucsc.edu/goldenpath/hg19/encodeDCC/wgEncodeAwgTfbsHaibHelas3Pol2Pcr1xUniPk.narrowPeak.gz] and K562; CAGE expression data [http://fantom.gsc.riken.jp/5/datafiles/latest/extra/CAGE_peaks/hg19.cage_peak_phase1and2combined_tpm_ann.osc.txt.gz]; GTEx V7 data [https://storage.googleapis.com/gtex_analysis_v7/single_tissue_eqtl_data/GTEx_Analysis_v7_eQTL.tar.gz]; ClinVar vcf file [ftp://ftp.ncbi.nlm.nih.gov/pub/clinvar/]. CTR-seq data were deposited on DNA Data Bank of Japan Sequencing Read Archive (accession number: DRA010491). The mapping computational pipeline used a prototype of primer-chop available at https://gitlab.com/mcfrith/primer-chop. The precise methods and command lines are provided as Supplementary Methods.

## Code availability
Data, alongside source code of the models, a readme.txt file and other instructions for installing and running the analyses are available on our git repository [https://gite.lirmm.fr/ibc/deepSTR]. This repository can be downloaded using the following command line:

```
curl https://gite.lirmm.fr/ibc/deepSTR/-/archive/master/deepSTR-master.zip --output DeepSTR.zip
```
or simply at [https://gite.lirmm.fr/ibc/deepSTR/-/archive/master/deepSTR-master.zip].

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

## Acknowledgements

We thank Cédric Notredame, Anthony Mathelier, Oriol Fornes Crespo, Philip Richmond, Jean-Christophe Andrau, Diego Garrido Martin, Dimitri D. Pervouchine, Roderic Guigo, Charles Plessy, and Chung Hon for their help in analyzing the data and for insightful suggestions. We also thank Takahiro Arakawa for the preparation and provision of cell culture samples. We are indebted to the researchers around the globe who generated experimental data and made them freely available. C.-H.L. is grateful to Marc Piechaczyk and Edouard Bertrand for their continued support. The work was supported by funding from CNRS (International Associated Laboratory "miREGEN"), INSERM-ITMO Cancer project "LIONS" BIO2015-04, *Plan d'Investissement d'Avenir* #ANR-11-BINF-0002 *Institut de Biologie Computationnelle* (young investigator grant to C-H.L.) and GEM Flagship project funded from Labex NUMEV (ANR-10-LABX-0020). M.G. was supported by a *Conventions Industrielles de Formation par la Recherche* (CIFRE) PhD fellowship from SANOFI R&D. FANTOM5 was made possible by the following grants: Research Grant for RIKEN Omics Science Center from MEXT to Y.H.; Grant of the Innovative Cell Biology by Innovative Technology (Cell Innovation Program) from the MEXT to Y.H.; Research Grant from MEXT to the RIKEN Center for Life Science Technologies; Research Grant to RIKEN Preventive Medicine and Diagnosis Innovation Program from MEXT to Y.H. This work was further supported by a Research Grant from MEXT to the RIKEN Center for Integrative Medical Sciences.

## Author contributions

C.B., M.S., M.G., C.M., W.W.W., M.d.H., L.B., and C.-H.L. analyzed and interpreted the data. M.S. and M.G. developed CNN models and studied the impact of ClinVar variants. J.R., Y.H., A.H., H.S., S.N., and I.M. generated CAGE data used in this study. M.d.H., J.S., and C.-H.L. generated Zenbu tracks. M.d.H. and C.-H.L. studied G bias at ENCODE read 5' ends. M.T., M.M., M.K.-I., S.N., S.N., T.K., H.N., and M.F. developed CTR-seq and generated data used in this study. Y.H., P.C., C.C., W.W.W., L.B., and C.-H.L. acquired fundings. C.-H.L. wrote the manuscript. All authors have read and approved the manuscript.

## Competing interests

The authors declare no competing interests.

## Additional information

## FANTOM consortium

Imad Abugessaisa[11], Stuart Aitken[12], Bronwen L. Aken[13,14], Intikhab Alam[15], Tanvir Alam[15], Rami Alasiri[16], Ahmad M. N. Alhendi[17], Hamid Alinejad-Rokny[18], Mariano J. Alvarez[19], Robin Andersson[20,21], Takahiro Arakawa[11,22], Marito Araki[23], Taly Arbel[24], John Archer[15], Alan L. Archibald[25], Erik Arner[11,22], Peter Arner[26], Kiyoshi Asai[8,27,28], Haitham Ashoor[15], Gaby Astrom[26], Magda Babina[29], J. Kenneth Baillie[25], Vladimir B. Bajic[15], Archana Bajpai[5], Sarah Baker[12], Richard M. Baldarelli[30], Adam Balic[25], Mukesh Bansal[19], Arsen O. Batagov[31], Serafim Batzoglou[32], Anthony G. Beckhouse[33], Antonio P. Beltrami[34], Carlo A. Beltrami[34], Nicolas Bertin[11,22,35], Sharmodeep Bhattacharya[24,36], Peter J. Bickel[24], Judith A. Blake[30], Mathieu Blanchette[37], Beatrice Bodega[38], Alessandro Bonetti[11,22], Hidemasa Bono[39], Jette Bornholdt[20,21], Michael Bttcher[11], Salim Bougouffa[15], Mette Boyd[20,21], Jeremie Breda[40,41], Frank Brombacher[42,43], James B. Brown[24,44], Carol J. Bult[30], A. Maxwell Burroughs[11,22,45], Dave W. Burt[25], Annika Busch[5], Giulia Caglio[46], Andrea Califano[19], Christopher J. Cameron[37], Carlo V. Cannistraci[47], Alessandra Carbone[48], Ailsa J. Carlisle[25], Piero Carninci[11,22], Kim W. Carter[49], Daniela Cesselli[34], Jen-Chien Chang[11], Julie C. Chen[50,51], Yun Chen[20,21], Marco Chierici[52], John Christodoulou[53], Yari Ciani[54], Emily L. Clark[25], Mehmet Coskun[20,55], Maria Dalby[20], Emiliano Dalla[54], Carsten O. Daub[22], Carrie A. Davis[56], Michiel J. L. de Hoon[11,22], Derek de Rie[11,57], Elena Denisenko[58], Bart Deplancke[59], Michael Detmar[60], Ruslan Deviatiiarov[11,61], Diego Di Bernardo[62], Alexander D. Diehl[63], Lothar C. Dieterich[60], Emmanuel Dimont[64], Sarah Djebali[65], Taeko Dohi[5,66], Jose Dostie[16], Finn Drablos[67],

Albert S. B. Edge[68], Matthias Edinger[69,70], Anna Ehrlund[26], Karl Ekwall[71], Arne Elofsson[72], Mitsuhiro Endoh[5], Hideki Enomoto[73], Saaya Enomoto[11], Mohammad Faghihi[74], Michela Fagiolini[75], Mary C. Farach-Carson[68,76,77,78], Geoffrey J. Faulkner[79], Alexander Favorov[80,81], Ana Miguel Fernandes[46], Carmelo Ferrai[46,82], Alistair R. R. Forrest[11,18,22], Lesley M. Forrester[25], Mattias Forsberg[83], Alexandre Fort[11,22], Margherita Francescatto[52], Tom C. Freeman[25], Martin Frith[27,84], Shinji Fukuda[5], Manabu Funayama[85], Cesare Furlanello[52], Masaaki Furuno[11,22], Chikara Furusawa[86,87], Hui Gao[71], Iveta Gazova[25], Claudia Gebhard[69,70], Florian Geier[88], Teunis B. H. Geijtenbeek[89], Samik Ghosh[5,90], Yanal Ghosheh[91], Thomas R. Gingeras[56], Takashi Gojobori[15,91], Tatyana Goldberg[92], Daniel Goldowitz[50], Julian Gough[93], Dario Greco[94], Andreas J. Gruber[40,41], Sven Guhl[29], Roderic Guigo[65], Reto Guler[42,43], Oleg Gusev[6,11,61], Stefano Gustincich[95,96], Thomas J. Ha[50], Vanja Haberle[97,98], Paul Hale[30], Bjrn M. Hallstrom[83,99], Michiaki Hamada[8,27,100], Lusy Handoko[11], Mitsuko Hara[101], Matthias Harbers[11,22], Jennifer Harrow[14], Jayson Harshbarger[11,22], Takeshi Hase[5], Akira Hasegawa[11,22], Kosuke Hashimoto[11,22], Taku Hatano[102], Nobutaka Hattori[85,102,103,104], Ryuhei Hayashi[105,106], Yoshihide Hayashizaki[6,22], Meenhard Herlyn[107], Peter Heutink[108], Winston Hide[64,109], Kelly J. Hitchens[110], Shannon Ho Sui[64], Peter A. C. 't Hoen[111], Chung Chau Hon[11], Fumi Hori[11,22], Masafumi Horie[112], Katsuhisa Horimoto[113], Paul Horton[8,27,114], Rui Hou[18], Edward Huang[115,116], Yi Huang[11], Richard Hugues[48], David Hume[25], Hans Ienasescu[20,21], Kei Iida[117,118], Tomokatsu Ikawa[5], Toshimichi Ikemura[119], Kazuho Ikeo[120], Norihiko Inoue[5], Yuri Ishizu[11,22], Yosuke Ito[6], Masayoshi Itoh[6,11,22], Anna V. Ivshina[31], Boris R. Jankovic[91], Piroon Jenjaroenpun[31], Rory Johnson[65], Mette Jorgensen[20,21], Hadi Jorjani[40,41], Anagha Joshi[25], Giuseppe Jurman[52], Bogumil Kaczkowski[11,22], Chieko Kai[121], Kaoru Kaida[11,22], Kazuhiro Kajiyama[11,22], Rajaram Kaliyaperumal[111], Eli Kaminuma[120], Takashi Kanaya[5], Hiroshi Kaneda[122], Philip Kapranov[123,124], Artem S. Kasianov[80,125], Takeya Kasukawa[11], Toshiaki Katayama[39], Sachi Kato[11,22], Shuji Kawaguchi[117], Jun Kawai[6,22], Hideya Kawaji[6,11,22], Hiroshi Kawamoto[5], Yuki I. Kawamura[66], Satoshi Kawasaki[126], Tsugumi Kawashima[11,22], Judith S. Kempfle[68], Tony J. Kenna[127], Juha Kere[71,128,129], Levon Khachigian[17], Hisanori Kiryu[130], Mami Kishima[11,22], Hiroyuki Kitajima[131], Toshio Kitamura[132,133], Hiroaki Kitano[5,90,134,135,136], Enio Klaric[54], Kjetil Klepper[67], S. Peter Klinken[18], Edda Kloppmann[92], Alan J. Knox[137], Yuichi Kodama[120], Yasushi Kogo[6], Miki Kojima[11,22], Soichi Kojima[101], Norio Komatsu[138], Hiromitsu Komiyama[139], Tsukasa Kono[11,22], Haruhiko Koseki[5], Shigeo Koyasu[5,140], Anton Kratz[11,22], Alexander Kukalev[46], Ivan Kulakovskiy[80,141,142], Anshul Kundaje[32,143], Hiroshi Kunikata[144,145], Richard Kuo[25], Tony Kuo[27], Shigehiro Kuraku[101], Vladimir A. Kuznetsov[31], Tae Jun Kwon[11,22], Matt Larouche[50], Timo Lassmann[11,22,49], Andy Law[25], Kim-Anh Le-Cao[127], Charles-Henri Lecellier[50,146], Weonju Lee[147], Boris Lenhard[97], Andreas Lennartsson[71], Kang Li[20,21,148], Ruohan Li[18], Berit Lilje[20,21], Leonard Lipovich[149,150], Marina Lizio[11,22], Gonzalo Lopez[19], Shigeyuki Magi[5], Gloria K. Mak[50], Vsevolod Makeev[80,141,151], Riichiro Manabe[11,22], Michiko Mandai[131], Jessica Mar[152], Kazuichi Maruyama[144], Taeko Maruyama[11], Elizabeth Mason[33], Anthony Mathelier[50], Hideo Matsuda[87], Yulia A. Medvedeva[80,153,154], Terrence F. Meehan[13], Niklas Mejhert[26], Alison Meynert[12], Norihisa Mikami[155], Akiko Minoda[11], Hisashi Miura[16,131], Yohei Miyagi[156], Atsushi Miyawaki[157], Yosuke Mizuno[158], Hiromasa Morikawa[155], Mitsuru Morimoto[131], Masaki Morioka[6], Soji Morishita[23], Kazuyo Moro[5,159], Efthymios Motakis[11,22], Hozumi Motohashi[160], Abdul Kadir Mukarram[71], Christine L. Mummery[161], Christopher J. Mungall[44], Yasuhiro Murakawa[6,11], Masami Muramatsu[158], Mitsuyoshi Murata[11,22], Kazunori Nagasaka[162], Takahide Nagase[112], Yutaka Nakachi[158], Fumio Nakahara[132,133], Kenta Nakai[163], Kumi Nakamura[11], Yasukazu Nakamura[120], Yukio Nakamura[164], Toru Nakazawa[144,145,165], Guy P. Nason[166], Chirag Nepal[98,167], Quan Hoang Nguyen[11], Lars K. Nielsen[33], Kohji Nishida[106], Koji M. Nishiguchi[144], Hiromi Nishiyori[11,22], Kazuhiro Nitta[11], Shuhei Noguchi[11], Shohei Noma[11,22], Cedric Notredame[65],

Soichi Ogishima[168], Naganari Ohkura[155,169], Hiroshi Ohno[5], Mitsuhiro Ohshima[170], Takashi Ohtsu[156], Yukinori Okada[5,155,171], Mariko Okada-Hatakeyama[5,172], Yasushi Okazaki[11,158], Per Oksvold[83], Valerio Orlando[91,173], Ghim Sion Ow[31], Mumin Ozturk[42,43], Mikhail Pachkov[40,41], Triantafyllos Paparountas[173], Suraj P. Parihar[42,43], Sung-Joon Park[163], Giovanni Pascarella[11,22], Robert Passier[161], Helena Persson[174], Ingrid H. Philippens[175], Silvano Piazza[54], Charles Plessy[11,22], Ana Pombo[46,82], Fredrik Ponten[176,177], Stéphane Poulain[11], Thomas M. Poulsen[27], Swati Pradhan[178,179,180], Carolina Prezioso[173], Clare Pridans[25], Xiang-Yang Qin[101], John Quackenbush[181,182], Owen Rackham[93,183], Jordan Ramilowski[11,22], Timothy Ravasi[91], Michael Rehli[69,70], Sarah Rennie[20], Tiago Rito[46], Patrizia Rizzu[108], Christelle Robert[25], Marco Roos[111], Burkhard Rost[92], Filip Roudnicky[60], Riti Roy[18], Morten B. Rye[67], Oxana Sachenkova[72], Pal Saetrom[67,184], Hyonmi Sai[185], Shinji Saiki[102], Mitsue Saito[185], Akira Saito[112,186], Shimon Sakaguchi[155,187], Mizuho Sakai[11,22], Saori Sakaue[5,171,188], Asako Sakaue-Sawano[157], Albin Sandelin[20,21], Hiromi Sano[11,22], Yuzuru Sasamoto[106], Hiroki Sato[121], Alka Saxena[22,189], Hideyuki Saya[190], Andrea Schafferhans[191], Sebastian Schmeier[58], Christian Schmidl[69], Daniel Schmocker[40,41], Claudio Schneider[34,54], Marcus Schueler[46], Erik A. Schultes[111], Gundula Schulze-Tanzil[192], Colin A. Semple[12], Shigeto Seno[87], Wooseok Seo[5], Jun Sese[27,193], Jessica Severin[11,22], Guojun Sheng[131,194], Jiantao Shi[64], Yishai Shimoni[19,195], Jay W. Shin[11,22], Javier SimonSanchez[108], Asa Sivertsson[83], Evelina Sjostedt[83,176], Cilla Soderhall[71], Georges St Laurent III[124,196], Marcus H. Stoiber[24,44], Daisuke Sugiyama[197], Kim M. Summers[25], Ana Maria Suzuki[11,22], Harukazu Suzuki[11,22], Kenji Suzuki[198], Mikiko Suzuki[199], Naoko Suzuki[11,22], Takahiro Suzuki[11,22], Douglas J. Swanson[50], Rolf K. Swoboda[107], Michihira Tagami[11,22], Ayumi Taguchi[162], Hazuki Takahashi[11,22], Masayo Takahashi[131], Kazuya Takamochi[198], Satoru Takeda[122], Yoichi Takenaka[87], Kin Tung Tam[18], Hiroshi Tanaka[168,200], Rica Tanaka[201], Yuji Tanaka[11,131,202], Dave Tang[11,22], Ichiro Taniuchi[5], Andrea Tanzer[65], Hiroshi Tarui[11,22], Martin S. Taylor[12], Aika Terada[28,84], Yasuhisa Terao[122], Alison C. Testa[18], Mark Thomas[14], Supat Thongjuea[11], Kentaro Tomii[27,28,84], Elena Torlai Triglia[46], Hiroo Toyoda[203], H. Gwen Tsang[25], Motokazu Tsujikawa[106], Mathias Uhlén[83], Eivind Valen[167], Marc van de Wetering[204], Erik van Nimwegen[40,41], Dmitry Velmeshev[74], Roberto Verardo[54], Morana Vitezic[20,21,22], Kristoffer Vitting-Seerup[20,21], Kalle von Feilitzen[83], Christian R. Voolstra[91], Ilya E. Vorontsov[80], Claes Wahlestedt[74], Wyeth W. Wasserman[50], Kazuhide Watanabe[11], Shoko Watanabe[11,22], Christine A. Wells[115,116], Louise N. Winteringham[18], Ernst Wolvetang[33], Haruka Yabukami[11,22], Ken Yagi[11], Takuji Yamada[205], Yoko Yamaguchi[206], Masayuki Yamamoto[207], Yasutomo Yamamoto[39], Yumiko Yamamoto[11,22], Yasunari Yamanaka[6], Kojiro Yano[208], Kayoko Yasuzawa[11], Yukiko Yatsuka[158], Masahiro Yo[164], Shunji Yokokura[144], Misako Yoneda[121], Emiko Yoshida[11], Yuki Yoshida[5], Masahito Yoshihara[11,106], Rachel Young[25], Robert S. Young[12], Nancy Y. Yu[71], Noriko Yumoto[5], Susan E. Zabierowski[209], Peter G. Zhang[50], Silvia Zucchelli[95,210] & Martin Zwahlen[83]

[11]Division of Genomic Technologies, RIKEN Center for Life Science Technologies, Yokohama, Japan. [12]MRC Human Genetics Unit, Institute of Genetics and Molecular Medicine, University of Edinburgh, Edinburgh, UK. [13]European Molecular Biology Laboratory, European Bioinformatics Institute, Wellcome Genome Campus, Hinxton, Cambridge, UK. [14]Wellcome Trust Sanger Institute, Wellcome Trust Genome Campus, Hinxton, UK. [15]Computational Bioscience Research Centre, King Abdullah University of Science and Technology (KAUST), Thuwal, Saudi Arabia. [16]Department of Biochemistry, McGill University, Montral, Qubec, Canada. [17]UNSW Centre for Vascular Research, University of New South Wales, Sydney, NSW, Australia. [18]Harry Perkins Institute of Medical Research, and the Centre for Medical Research, University of Western Australia, QEII Medical Centre, Perth, WA, Australia. [19]Department of Systems Biology, Columbia University Medical Center, Columbia University, New York, NY, USA. [20]The Bioinformatics Centre, Department of Biology, University of Copenhagen, Copenhagen, Denmark. [21]Biotech Research and Innovation Centre, University of Copenhagen, Copenhagen, Denmark. [22]RIKEN Omics Science Center (OSC), Yokohama, Japan. [23]Department of Transfusion Medicine and Stem Cell Regulation, Juntendo University Graduate School of Medicine, Tokyo, Japan. [24]Department of Statistics, University of California Berkeley, Berkeley, CA, USA. [25]The Roslin Institute and Royal (Dick) School of Veterinary Studies, University of Edinburgh, Easter Bush, UK. [26]Department of Medicine, Karolinska Institute at Karolinska University Hospital, Huddinge, Sweden. [27]Artificial Intelligence Research Center (AIRC), National Institute of Advanced Industrial Science and Technology (AIST), Tokyo, Japan. [28]Biotechnology Research Institute for Drug Discovery, National Institute of Advanced Industrial Science and Technology (AIST), Tokyo, Japan. [29]Department of Dermatology and Allergy, Charit Campus Mitte, Universitatsmedizin Berlin, Berlin, Germany. [30]The Jackson Laboratory, Bar Harbor, ME, USA. [31]Bioinformatics Institute, Agency for Science, Technology and Research (A*STAR), Singapore, Singapore. [32]Department of Computer Science, Stanford University, Stanford, CA, USA. [33]Australian Institute for Bioengineering and Nanotechnology (AIBN), University of Queensland, Brisbane St Lucia, QLD, Australia.

[34]Department of Medical and Biological Sciences, University of Udine, Udine, Italy. [35]Cancer Science Institute of Singapore, National University of Singapore, Singapore, Singapore. [36]Department of Statistics, Oregon State University, Corvallis, OR, USA. [37]McGill Centre for Bioinformatics and School of Computer Science, McGill University, Montral, Qubec, Canada. [38]Genome Biology Unit, Istituto Nazionale di Genetica Molecolare (INGM) 'Romeo and Enrica Invernizzi', Milan, Italy. [39]Database Center for Life Science, Research Organization of Information and Systems, Tokyo, Japan. [40]Biozentrum, University of Basel, Basel, Switzerland. [41]Swiss Institute of Bioinformatics, Basel, Switzerland. [42]International Centre for Genetic Engineering and Biotechnology, Cape Town Component, Cape Town, South Africa. [43]Division of Immunology, Institute of Infectious Diseases and Molecular Medicine, Health Science Faculty, University of Cape Town, Cape Town, South Africa. [44]Genomics Division, Lawrence Berkeley National Laboratory, Berkeley, CA, USA. [45]National Center for Biotechnology Information, National Library of Medicine, National Institutes of Health, Bethesda, MD, USA. [46]Berlin Institute for Medical Systems Biology, Max-Delbruck Centre for Molecular Medicine, Berlin, Germany. [47]Biotechnology Center, Technische Universitat Dresden, Dresden, Germany. [48]Sorbonne Universités, Université Pierre et Marie Curie, Laboratoire de Biologie Computationnelle et Quantitative, Paris, France. [49]Telethon Kids Institute, The University of Western Australia, Subiaco, WA, Australia. [50]Department of Medical Genetics, Centre for Molecular Medicine and Therapeutics, Child and Family Research Institute, University of British Columbia, Vancouver, British Columbia, Canada. [51]Graduate Program in Bioinformatics, University of British Columbia, Vancouver, British Columbia, Canada. [52]Fondazione Bruno Kessler, Trento, Italy. [53]Children's Hospital at Westmead, Sydney, NSW, Australia. [54]Laboratorio Nazionale Consorzio Italiano Biotecnologie (LNCIB), Trieste, Italy. [55]Department of Gastroenterology, Medical Section, Herlev Hospital, University of Copenhagen, Herlev, Denmark. [56]Functional Genomics, Cold Spring Harbor Laboratory, Cold Spring Harbor, NY, USA. [57]Centre for Integrative Bioinformatics (IBIVU), VU University Amsterdam, Amsterdam, The Netherlands. [58]Institute of Natural and Mathematical Sciences, Massey University Auckland, Albany, New Zealand. [59]Ecole Polytechnique Fdrale de Lausanne and Swiss Institute of Bioinformatics, Lausanne, Switzerland. [60]Institute of Pharmaceutical Sciences, Swiss Federal Institute of Technology, ETH Zurich, Zurich, Switzerland. [61]Institute of Fundamental Medicine and Biology, Kazan Federal University, Kazan, Russia. [62]Telethon Institute of Genetics and Medicine (TIGEM), Pozzuoli, Italy. [63]Department of Neurology, University at Buffalo School of Medicine and Biomedical Sciences, Buffalo, NY, USA. [64]Department of Biostatistics, Harvard School of Public Health, Boston, MA, USA. [65]Centre for Genomic Regulation (CRG), The Barcelona Institute of Science and Technology, Barcelona, Spain. [66]Department of Gastroenterology, Research Center for Hepatitis and Immunology, Research Institute, National Center for Global Health and Medicine, Chiba, Japan. [67]Department of Cancer Research and Molecular Medicine, Norwegian University of Science and Technology, Trondheim, Norway. [68]Department of Otology and Laryngology, Harvard Medical School, Boston, MA, USA. [69]Department of Internal Medicine III, University Hospital Regensburg, Regensburg, Germany. [70]Regensburg Centre for Interventional Immunology (RCI), Regensburg, Germany. [71]Department of Biosciences and Nutrition, Karolinska Institute, Stockholm, Sweden. [72]Department of Biochemistry and Biophysics, Stockholm University, Stockholm, Sweden. [73]Division of Neural Differentiation and Regeneration, Kobe University Graduate School of Medicine, Kobe, Japan. [74]Department of Psychiatry and Behavioral Sciences, University of Miami Miller School of Medicine, Miami, FL, USA. [75]F.M. Kirby Neurobiology Center, Department of Neurology, Boston Children's Hospital, Harvard Medical School, Boston, MA, USA. [76]Department of Biological Sciences, University of Delaware, Newark, DE, USA. [77]Department of Biochemistry and Cell Biology, Rice University, Houston, TX, USA. [78]Department of Bioengineering, Rice University, Houston, TX, USA. [79]Mater Research Institute, and Queensland Brain Institute, University of Queensland, Brisbane, QLD, Australia. [80]Vavilov Institute of General Genetics, Russian Academy of Sciences, Moscow, Russia. [81]Department of Oncology, Division of Biostatistics and Bioinformatics, Johns Hopkins University School of Medicine, Baltimore, MD, USA. [82]Genome Function Group, MRC Clinical Sciences Centre, Imperial College London, London, UK. [83]Science for Life Laboratory, KTH-Royal Institute of Technology, Stockholm, Sweden. [84]Department of Computational Biology and Medical Sciences, University of Tokyo, Tokyo, Japan. [85]Research Institute for Diseases of Old Age, Juntendo University Graduate School of Medicine, Tokyo, Japan. [86]RIKEN Quantitative Biology Center, Suita, Japan. [87]Graduate School of Information Science and Technology, Osaka University, Suita, Japan. [88]Department of Biomedicine, Bioinformatics Core Facility, University Hospital Basel, Basel, Switzerland. [89]Academic Medical Center, University of Amsterdam, Amsterdam, The Netherlands. [90]The Systems Biology Institute, Tokyo, Japan. [91]Division of Biological and Environmental Sciences & Engineering, King Abdullah University of Science and Technology (KAUST), Thuwal, Saudi Arabia. [92]Department for Bioinformatics and Computational Biology, Technische Universitłt Mnchen, Garching, Germany. [93]Department of Computer Science, University of Bristol, Bristol, UK. [94]Institute of Biotechnology, University of Helsinki, Helsinki, Finland. [95]Area of Neuroscience, International School for Advanced Studies (SISSA), Trieste, Italy. [96]Department of Neuroscience and Brain Technologies, Italian Institute of Technologies (IIT), Genoa, Italy. [97]Faculty of Medicine, Imperial College London, London, UK. [98]Department of Biology, University of Bergen, Bergen, Norway. [99]Department of Proteomics, KTH-Royal Institute of Technology, Stockholm, Sweden. [100]Department of Electrical Engineering and Bioscience, Faculty of Science and Engineering, Waseda University, Tokyo, Japan. [101]RIKEN Center for Life Science Technologies, Division of Bio-Function Dynamics Imaging, Kobe, Japan. [102]Department of Neurology, Juntendo University Graduate School of Medicine, Tokyo, Japan. [103]Department of Treatment and Research in Multiple Sclerosis and Neuro-intractable Disease, Juntendo University Graduate School of Medicine, Tokyo, Japan. [104]Department of Research for Parkinsons Disease, Juntendo University Graduate School of Medicine, Tokyo, Japan. [105]Department of Stem Cells and Applied Medicine, Osaka University Graduate School of Medicine, Suita, Japan. [106]Department of Ophthalmology, Osaka University Graduate School of Medicine, Suita, Japan. [107]Melanoma Research Center, The Wistar Institute, Philadelphia, PA, USA. [108]German Center for Neurodegenerative Diseases (DZNE), Tubingen, Germany. [109]Sheffield Institute for Translational Neuroscience, University of Sheffield, Sheffield, UK. [110]Australian Infectious Diseases Research Centre (AID), University of Queensland, Brisbane, QLD, Australia. [111]Department of Human Genetics, Leiden University Medical Center, Leiden, The Netherlands. [112]Department of Respiratory Medicine, Graduate School of Medicine, University of Tokyo, Tokyo, Japan. [113]Molecular Profiling Research Center for Drug Discovery, National Institute of Advanced Industrial Science and Technology (AIST), Tokyo, Japan. [114]Computational Biology Research Center, National Institute of Advanced Industrial Science and Technology (AIST), Tokyo, Japan. [115]The University of Melbourne Centre for Stem Cell Systems, School of Biomedical Sciences, The University of Melbourne, Victoria, Australia. [116]Walter and Eliza Hall Institute of Medical Research, Melbourne, VIC, Australia. [117]RIKEN Bioinformatics and Systems Engineering Division (BASE), Yokohama, Japan. [118]Medical Research Support Center, Kyoto University Graduate School of Medicine, Kyoto, Japan. [119]Department of Bioscience, Nagahama Institute of Bio-Science and Technology, Nagahama, Japan. [120]Center for Information Biology and DNA Data Bank of Japan, National Institute of Genetics, Mishima, Japan. [121]Laboratory Animal Research Center, Institute of Medical Science, University of Tokyo, Tokyo, Japan. [122]Department of Obstetrics and Gynecology, Juntendo University, Tokyo, Japan. [123]Institute of Genomics, School of Biomedical Sciences, Huaqiao University, Xiamen, China. [124]St. Laurent Institute, Woburn, MA, USA. [125]A.N. Belozersky Institute of Physico-Chemical Biology, Lomonosov Moscow State University, Moscow, Russia. [126]Department of Ophthalmology, Kyoto Prefectural University of Medicine, Kyoto, Japan. [127]Diamantina Institute, University of Queensland, Brisbane St Lucia, QLD, Australia. [128]Folkhalsan Institute of Genetics, Helsinki, Finland. [129]Science for Life Laboratory, Karolinska Institute, Solna, Sweden. [130]Department of Computational Biology, Faculty of Frontier Sciences, University of Tokyo, Chiba, Japan. [131]RIKEN Center for Developmental Biology, Kobe, Japan. [132]Division of Cellular Therapy, Institute of Medical Science, University of Tokyo, Tokyo, Japan. [133]Division of Stem Cell Signaling, Institute of Medical Science, University of Tokyo, Tokyo, Japan. [134]Sony Computer Science Laboratories, Inc, Tokyo, Japan. [135]Systems Biology Institute (SBI) Australia, Monash University,

Clayton, VIC, Australia. [136]Okinawa Institute of Science and Technology, Onna, Japan. [137]Department of Respiratory Medicine and Nottingham Respiratory Research Unit, University of Nottingham, Nottingham, UK. [138]Department of Hematology, Juntendo University Graduate School of Medicine, Tokyo, Japan. [139]Department of Coloproctological Surgery, Faculty of Medicine, Juntendo University School of Medicine, Tokyo, Japan. [140]Department of Microbiology and Immunology, Keio University School of Medicine, Tokyo, Japan. [141]Engelhardt Institute of Molecular Biology, Russian Academy of Sciences, Moscow, Russia. [142]Skolkovo Institute of Science and Technology, Moscow, Russia. [143]Department of Genetics, Stanford University, Stanford, CA, USA. [144]Department of Ophthalmology and Visual Science, Tohoku University Graduate School of Medicine, Sendai, Japan. [145]Department of Retinal Disease Control, Tohoku University Graduate School of Medicine, Sendai, Japan. [146]Institute of Molecular Genetics of Montpellier, Montpellier, France. [147]Department of Dermatology, Kyungpook National University School of Medicine, Daegu, South Korea. [148]Department of Mathematical Sciences, University of Copenhagen, Copenhagen, Denmark. [149]Center for Molecular Medicine and Genetics, Wayne State University, Detroit, MI, USA. [150]Department of Neurology, School of Medicine, Wayne State University, Detroit, MI, USA. [151]Department of Medical and Biological Physics, Moscow Institute of Physics and Technology, Moscow, Russia. [152]Department of Systems and Computational Biology, Albert Einstein College of Medicine, New York, NY, USA. [153]IMPPC, Institute of Predictive and Personalized Medicine of Cancer, Badalona, Spain. [154]Institute of Bioengineering, Research Center of Biotechnology, Moscow, Russia. [155]Immunology Frontier Research Center, Osaka University, Suita, Japan. [156]Kanagawa Cancer Center Research Institute, Yokohama, Japan. [157]RIKEN Brain Science Institute, Saitama, Japan. [158]Research Center for Genomic Medicine, Saitama Medical University, Saitama, Japan. [159]Department of Medical Life Science, Graduate School of Medical Life Science, Yokohama City University, Yokohama, Japan. [160]Department of Gene Expression Regulation, Institute of Development, Aging and Cancer, Tohoku University, Sendai, Japan. [161]Department of Anatomy and Embryology, Leiden University Medical Center, Leiden, The Netherlands. [162]Department of Obstetrics and Gynecology, Graduate School of Medicine, University of Tokyo, Tokyo, Japan. [163]Human Genome Center, The Institute of Medical Science, University of Tokyo, Tokyo, Japan. [164]RIKEN BioResource Center, Tsukuba, Japan. [165]Department of Advanced Ophthalmic Medicine, Tohoku University Graduate School of Medicine, Sendai, Japan. [166]School of Mathematics, University of Bristol, Bristol, UK. [167]Department of Informatics, University of Bergen, Bergen, Norway. [168]Tohoku Medical Megabank Organization, Tohoku University, Sendai, Japan. [169]Department of Frontier Research in Tumor Immunology, Center of Medical Innovation and Translational Research, Osaka University, Osaka, Japan. [170]Department of Biochemistry, Ohu University School of Pharmaceutical Sciences, Koriyama, Japan. [171]Department of Statistical Genetics, Osaka University Graduate School of Medicine, Suita, Japan. [172]Institute for Protein Research, Osaka University, Suita, Japan. [173]Dulbecco Telethon Institute at IRCSS Fondazione Santa Lucia, Rome, Italy. [174]Division of Oncology and Pathology, Department of Clinical Sciences, Lund University, Lund, Sweden. [175]Department of Immunobiology, Biomedical Primate Research Centre, Rijswijk, The Netherlands. [176]Department of Immunology, Genetics and Pathology, Uppsala University, Uppsala, Sweden. [177]Science for Life Laboratory, Uppsala University, Uppsala, Sweden. [178]Department of BioSciences, Rice University, Houston, TX, USA. [179]Center for Translational Cancer Research, Helen F. Graham Cancer Center & Research Institute, Newark, DE, USA. [180]Department of Biomedical Engineering, University of Delaware, Newark, DE, USA. [181]Department of Biostatistics and Computational Biology, Dana-Farber Cancer Institute and Harvard Medical School, Boston, MA, USA. [182]Department of Biostatistics, Harvard T.H. Chan School of Public Health, Boston, MA, USA. [183]Program in Cardiovascular and Metabolic Disorders, DukeNUS Medical School, Singapore, Singapore. [184]Department of Computer and Information Science, Norwegian University of Science and Technology, Trondheim, Norway. [185]Division of Breast Oncology, Juntendo University School of Medicine, Tokyo, Japan. [186]Division for Health Service Promotion, University of Tokyo, Tokyo, Japan. [187]Department of Experimental Pathology, Institute for Frontier Medical Sciences, Kyoto University, Kyoto, Japan. [188]Department of Allergy and Rheumatology, Graduate School of Medicine, University of Tokyo, Tokyo, Japan. [189]Biomedical Research Centre at Guy's and St Thomas' Trust, Genomics Core Facility, Guy's Hospital, London, UK. [190]Division of Gene Regulation, Institute for Advanced Medical Research, Keio University School of Medicine, Tokyo, Japan. [191]Department of Informatics, Technische Universität Mnchen, Garching, Germany. [192]Paracelsus Medical University, Institute of Anatomy, Nuremberg, Germany. [193]Department of Computer Science, Tokyo Institute of Technology, Tokyo, Japan. [194]International Research Center for Medical Sciences, Kumamoto University, Kumamoto, Japan. [195]Department of Neurology and Center for Translational Systems Biology, Mount Sinai School of Medicine, New York, NY, USA. [196]Department of Molecular Biology, Cell Biology, and Biochemistry, Brown University, Providence, RI, USA. [197]Department of Research and Development of Next Generation Medicine, Faculty of Medical Sciences, Kyushu University, Fukuoka, Japan. [198]Department of General Thoracic Surgery, Juntendo University School of Medicine, Tokyo, Japan. [199]Center for Radioisotope Sciences, Tohoku University Graduate School of Medicine, Sendai, Japan. [200]Department of Systems Biology, Graduate School of Biochemical Science, Tokyo Medical and Dental University, Tokyo, Japan. [201]Department of Plastic and Reconstructive Surgery, Juntendo University Graduate School of Medicine, Tokyo, Japan. [202]RIKEN Advanced Center for Computing and Communication, Preventive Medicine and Applied Genomics Unit, Yokohama, Japan. [203]Department of Clinical Molecular Genetics, School of Pharmacy, Tokyo University of Pharmacy and Life Sciences, Tokyo, Japan. [204]Hubrecht Institute, Utrecht, The Netherlands. [205]Department of Biological Information, Graduate School of Bioscience and Biotechnology, Tokyo Institute of Technology, Tokyo, Japan. [206]Department of Biochemistry, Nihon University School of Dentistry, Tokyo, Japan. [207]Graduate School of Medicine, Tohoku University, Sendai, Japan. [208]Faculty of Information Science and Technology, Osaka Institute of Technology, Hirakata, Japan. [209]The SKI Stem Cell Research Facility, The Center for Stem Cell Biology and Developmental Biology Program, Sloan Kettering Institute, New York, NY, USA. [210]Department of Health Sciences, Universit del Piemonte Orientale, Novara, Italy.

