## [Peer Review File · Nature Communications]

Reviewers' Comments:

Reviewer #1:

Remarks to the Author:

Major comments:

(1) The authors found that 89,948 CAGE peaks ($\approx 8.6\%$) initiate at 84,555 STRs... is this more so than expected by chance or just the rate one would observe for 84,555 random features?

(2) The authors refer to the sequence level instructions for the STR transcription. This is one of the most interesting findings, but still underdeveloped. While conservation appears to associate, the authors never demonstrate these conserved regions are in fact instructions. Currently they have observed a pattern, which does not imply it causes (or instructs).

(3) Its unclear whether the regions identified are any more clinically relevant than other features. If one were to take an equivalent quantity quantity of randomly generated regions compared to STRs, perhaps they would also identify 34,578 of those random regions harbour at least one ClinVar variants. This is another example of a section in the manuscript where numbers are provided, but no statistics to put the findings into context.

(4) The authors could further develop associations of STRs with different biotypes -- ie, are there different patterns between STRs and non-coding RNAs, enhancer RNAs, etc.

(5) Overall, its not clear from this manuscript what the critical limitation to this point is for identifying different STR classes. Why was this not possible before since one could observe their DNA repeat motif? Further, while this CNN model is relevant for analyzing CAGE data, its unclear how broadly applicable this is since there are various strategies for monitoring transcription.

(6) This study would be more impactful had the authors demonstrated the application of the CNN models for improving eSTR computations.

Minor comments:

(1) There are numerous grammatical errors throughout the manuscript. For instance:

"This type of machine learning approches takes as input the DNA" should read "This type of machine learning approach takes as input the DNA". And "making hard to learn models" should be "making it hard to learn models".

Reviewer #2:

Remarks to the Author:

Short tandem repeats (STRs) are polymorphic genetic elements associated with gene expression and disease. STRs are also transcribed, and the function of these STR-RNAs only begins to be studied. Here, Grapotte et al. conduct a systematic investigation of transcribed STRs by overlapping STRs with genome-wide maps of transcription start sites obtained by CAGE mapping. They furthermore develop and apply a new assay, Cap Trap RNA-seq, to validate these transcription sites. Next, the authors train machine learning models to predict different classes of transcribed STRs using their flanking sequence. These results indicate that transcription of STRs could be influenced by flanking regions in a way that is conserved between human and mouse and that differ between STR classes.

This study is welcome because systematic studies of transcription initiation at STRs have been lacking. However, I have major concerns concerning i) the delineation of the contribution compared to the state of current knowledge, ii) the lack of insights, and iii) inconsistencies casting doubt about the rigour of the analysis. Regarding the writing style, the manuscript is in parts quite hard to read because of very complex sentence structures. It also contains a large number of typos. Major and minor concerns are detailed below. Altogether, I suggest a major revision.

Major

1. The introduction omits prior studies on STR-containing RNAs. It gives the false impression that this is the first study describing transcribed STRs ("we hypothesized that transcription initiation also occurs at STRs"). The introduction of Mills et al eLife, 2019 (<https://elifesciences.org/articles/48940>; not cited) provides several references that should be mentioned. (Yap et al., 2018; Jain and Vale, 2017; Zhu et al., 2011; McNulty et al., 2017; Johnson et al., 2017; Velazquez Camacho et al., 2017; Shirai et al., 2017; Rošić et al., 2014). The authors should thereby precisely delineate the novelty of their work with respect to this prior work.
2. The authors report that 89,948 CAGE initiate at 84,555 STRs. Is that overlap statistically significant? This depends on how overlap is computed. Are the authors using some windows round peaks and if so, how large are they?
3. The (T)_n class represents the same class than the (A)_n. Is that the reason why these classes have equal sizes in Fig 1B? But then, why do the fractions differ in 1C?
4. I am confused by the fact that the model uses the "3' end of each STR". STRs are determined using an algorithm run only on the + strand. How is the 3' end of an STR defined? What is the biological interpretation of it? Why using the 3' end for modeling transcription initiation? I would expect focusing on the promoter, i.e. toward the 5' end and making use of CAGE to define the strand of an STR. More justifications would help.
5. The authors state "the discovery that STR flanking sequences are not inert but rather contain important features that play critical roles in their biology". This is not exactly novel. Actually, I found the insights of this work to be quite shallow despite previous work on the topic. Notably, Sun et al., Cell, 2018 (although cited by the authors) showed that many disease-associated STRs (daSTRs) are located at boundaries demarcating 3D chromatin domains. Would the DL models improve with chromatin domain annotation? Do the authors find chromatin boundary motifs such as CTCF to be predictive?
6. Model performance is not compared to any baseline model. What is the performance of a predictor that returns the median CAGE count for each STR class?
7. Major aspects of the CNN training procedure are not provided. These include: How did the training data was stratified? What are STR class counts in the training and testing data? How many STR's have a CAGE-peak in the training and testing data?
8. Establishing proper cross-validation in sequence-based predictions can be difficult due to homologous regions. Some of STRS may be part of homologous regions. Deep models may then overfit. The author should assess the issue of homologous regions.
9. The input and target of the CNN should be precisely described. A figure showing how a STR sequence + flanking regions is converted into an input matrix would be very helpful for understanding. Moreover, it is not clear which parts of the sequence exactly were replaced with "N".
10. A density-scatter or hexbin plots showing predicted vs observed CAGE counts on held out data should be provided to visualize the quality of the predictions.
11. I missed some model interpretation. Can model interpretation technique (see review Eraslan et al. Nat Review Genetics 2019) indicate TF binding sites? For directionality, does the model capture known motifs involved in transcription directionality (e.g. U1 binding sites Almada et al Nature 2013, probably motifs have been reported since then) ?
12. On page 10, it is stated that 766,747 (T)_n elements are predicted. However, in Figure 1B there are only ~ 400,000 T(n) STR's. Where does this difference stem from?

13. Figure 5, 6, and S8 should be provided for metrics computed on held-out test data.
14. Page 12. The statement "formally demonstrating the existence of STR class-specific features for transcription prediction." is too strong. One cannot exclude that one have obtained different results, would other algorithms or mathematical functions have been applied.
15. Figure 7 should be provided for ClinVar pathogenic and ClinVar benign variants separately (side-by-side) to assess whether there is an enrichment for pathogenic variants. The authors make a strong claim in this direction in the abstract but it is not supported without such comparison.
16. Fig 7B needs a multiple-testing corrected statistical test to establish that the enrichment shown around the TSS is significant.
17. Page 15, this statement is unclear: "Likewise, several diseases were found enriched comparing variant fractions located at transcribed STRs (Fisher's exact test $< 5e-3$, Supplementary Table S2)". What is meant with "variant fraction" and what is meant by "enriched"? I could not reconstruct the Fisher test contingency tables leading to the p-values provided in Suppl. Table S2. Also, multiple testing correction on this large number of tests should be applied.
18. Page 16 states that the models have learned key positions that correspond precisely to genetic variants linked to human diseases. However, it is stated on page 15: "The clinical significance of the variants, as defined in the ClinVar database, does not appear directly linked to the transcription rate of STRs". How do these statements connect?
19. As this claim is provided in the abstract, it should be very precisely nailed or removed.

Minor

20. Some of the other catalogs mentioned on page 5 ("compared to other catalogs.") should be explicitly names and referred to.
21. Figure 3: To directly compare values in 3A and 3B, a scatter plot of the values in 3B against those in 3A would be more informative (labelling points with STR names). Also, showing confidence intervals (e.g. binomial) would be useful. An alternative to the scatterplot would be side-by-side bar plots (With confidence intervals).
22. Page 9: "as defined in [4]," does not read well. A better style would be "as defined by <name et al> [4],".
23. Page 9, typo: "We used deep" -> "We used a deep"
24. Page 9. It is written that "Transcription at (A)_n, which is mostly detected on the (-) strand, does confirm the observation that transcription at (T)_n is mostly (+)." This sentence suggests that (T)_n is only transcribed on the (+) strand. But what it actually shows is that (T)_n is transcribed in the sense direction of whatever strand it is on, whereas (A)_n is not. That is, the strand where (T)_n is found (the complement of (A)_n) is the one that is transcribed.
25. Page 10, typo: "approche" -> "approaches"
26. Page 11, typo: "we masked the 7bases located downstream the STR 3' ends" -> "we masked the 7 bases located downstream of the STR 3' ends"

27. Page 13. "compare Figure 1B and Figure 6A". A scatter-plot should be provided to facilitate this comparison, with labelled dots. The R package ggrepel offers nice options for dot labelling.
28. Page 13 "Less reliable than the human one" should be "less reliable than the human one" (than)
29. The authors could discuss the observation that STR transcripts mostly reside inside the nucleus and do not get exported.
30. Page 17 "the findings made by by Bertuzzi et al. in": twice "by".
31. Usage of the model: Could the authors describe how the model can be used by practitioners wanting to, for example, interpret variants on patient data. Is there a simple workflow to apply the model to standard bioinformatics files such as variants in a vcf, or must the user manually transform the input data into a model with specific input format?
32. Fig 7, "variance". Do you mean "variants"?
33. The discussion is very repetitive with respect to the introduction.
34. On page 26, what does "brut force algorithms" exactly mean? Did the authors try every possible numeric value to find the optimal result? Moreover, the correct spelling is "brute force".

Reviewer #1:

Major comments:

(1) The authors found that 89,948 CAGE peaks (← 8.6%) initiate at 84,555 STRs... is this more so than expected by chance or just the rate one would observe for 84,555 random features?

We have generated a set of 1,620,030 randomly chosen intervals using bedtools shuffle. Only 2.3% of these intervals intersected with CAGE peaks, indicating that the percentage (8.6%) observed in the case of STRs is significant (Fisher's exact test p-value < 2.2e-16). All details of the analysis have been added in the revised version of the manuscript (see Results, page 5, and Methods sections, page 24)

(2) The authors refer to the sequence level instructions for the STR transcription. This is one of the most interesting findings, but still underdeveloped. While conservation appears to associate, the authors never demonstrate these conserved regions are in fact instructions. Currently they have observed a pattern, which does not imply it causes (or instructs).

We acknowledge that machine learning approaches only unveil correlation between predictive and predicted features, not direct causation. One way to clarify that point, in our case, is to assess whether modifying DNA sequence (i.e. predictive features) with genetic variants truly impacts transcription initiation. We sought to tackle this problem by looking at TSSs harboring variants acting as eQTLs for the corresponding genes, in a scenario similar to that described by Bertuzzi *et al.* in the case of a minisatellite [reference #20].

Details of the analyses and results are provided in the revised manuscript (Supplementary Figures S16 and S17 and page 17-19). Our idea was to compare the sign of the difference of the predictions made by our models for the reference and the alternative alleles and the sign of the eQTL slope (i.e. gene expression increase (slope > 0) or decrease (slope < 0)). We now show that, when the predictions are accurate on the reference genome (error <= 0.2), the models are able to predict the impact on expression i.e. in most cases, the sign of the difference between the predictions made with the alternative and predictive alleles is similar to that of the eQTL slope. Importantly, this is no longer observed for the STRs where the models perform poorly on the reference genome (error > 0.2). Binomial tests were used to statistically assess the relevance of these findings. Thus, when accurate, our models are able to predict the effects of eQTLs, supporting a causal relationship between the predictive and the predicted variables rather than a mere correlation. We also changed the term 'instructions' for 'features' throughout the revised manuscript before showing these results.

(3) Its unclear whether the regions identified are any more clinically relevant than other features. If one were to take an equivalent quantity quantity of randomly generated regions compared to STRs, perhaps they would also identify 34,578 of those random regions harbour at least one ClinVar variants. This is another example of a section in the manuscript where numbers are provided, but no statistics to put the findings into context.

When considering a set of 3,076,234 randomly chosen intervals (3,076,234 being the number of STRs with strand orientation thanks to CAGE data), we found 53,679 variants intersecting with these intervals, even indicative of a depletion of ClinVar variants at STRs (1.7% for random vs. 1.1% for STRs, Fisher's exact test p-value < 2.2e-16). The number '34,578' was initially indicated only to support the results depicted in Figure 7A. We have now modified the manuscript to make this aspect clearer (page 16).

Please see also Reviewer #2 point #15

(4) The authors could further develop associations of STRs with different biotypes -- ie, are there different patterns between STRs and non-coding RNAs, enhancer RNAs, etc.

As suggested by Reviewer#1, we have computed the enrichment of STR classes in FANTOM CAT biotypes and the computations are now provided as Supplementary Table S3 (page 20). The strongest enrichments correspond to $(A)_n$, $(AT)_n$ and $(AAAT)_n$ at enhancers, which are GC-poor sequences compared to promoters for instance [reference #50].

(5) Overall, its not clear from this manuscript what the critical limitation to this point is for identifying different STR classes. Why was this not possible before since one could observe their DNA repeat motif? Further, while this CNN model is relevant for analyzing CAGE data, its unclear how broadly applicable this is since there are various strategies for monitoring transcription.

We acknowledge that STR classes are defined by their DNA repeat motif. This is indeed the definition used in our study (see Introduction section page 4). Rather, Figure 5B shows that this classification is also possible when considering 50bp STR flanking sequences only, and masking the DNA repeat motif. This point has now been clearly stated in the revised manuscript (page 11).

We agree with Reviewer#1 that many methods exist to monitor transcription initiation. Our models were optimized to predict CAGE signal and cannot, as such, be directly applied to other types of data. However the methodology used here is generic and could be applied to other type of data as long as we can associate a numeric signal to a specific genomic region. This limitation has clearly been indicated in the revised manuscript (page 31).

(6) This study would be more impactful had the authors demonstrated the application of the CNN models for improving eSTR computations.

Several eQTLs considered in our response to point#2 implicate variants located in STR flanking sequences that do not affect their length. However, our results show that these eQTLs represent in fact genuine eSTRs, which were not considered in previous eSTR computations. As such, we believe that these results demonstrate that the application of CNN models can indeed improve eSTR computations simply allowing considering more variants, in particular in STR flanking sequences and re-assigning eQTLs as eSTRs. We have included this point in the Discussion section of the revised manuscript (page 21).

We are now in the process of computing eSTRs considering variants in flanking sequences at a genome-wide scale using GTEx data. Given the time and amount of work required to process these data, we believe these analyses could form the body of a completely separate study. Not providing these genome-wide analyses would probably not impinge the message conveyed by our study, which is aimed at describing the discovery of a predictable transcription initiation at STRs.

Minor comments:

(1) There are numerous grammatical errors throughout the manuscript. For instance: "This type of machine learning approches takes as input the DNA" should read "This type of machine learning approach takes as input the DNA". And "making hard to learn models" should be "making it hard to learn models".

We have corrected the manuscript accordingly.

Reviewer #2 (Remarks to the Author):

Short tandem repeats (STRs) are polymorphic genetic elements associated with gene expression and disease. STRs are also transcribed, and the function of these STR-RNAs only begins to be studied. Here, Grapotte et al. conduct a systematic investigation of transcribed STRs by overlapping STRs with genome-wide maps of transcription start sites obtained by CAGE mapping. They furthermore develop and apply a new assay, Cap Trap RNA-seq, to validate these transcription sites. Next, the authors train machine learning models to predict different classes of transcribed STRs using their flanking sequence. These results indicate that transcription of STRs could be influenced by flanking regions in a way that is conserved between human and mouse and that differ between STR classes.

This study is welcome because systematic studies of transcription initiation at STRs have been lacking. However, I have major concerns concerning i) the delineation of the contribution compared to the state of current knowledge, ii) the lack of insights, and iii) inconsistencies casting doubt about the rigour of the analysis. Regarding the writing style, the manuscript is in parts quite hard to read because of very complex sentence structures. It also contains a large number of typos. Major and minor concerns are detailed below. Altogether, I suggest a major revision.

Major

1. The introduction omits prior studies on STR-containing RNAs. It gives the false impression that this is the first study describing transcribed STRs (“we hypothesized that transcription initiation also occurs at STRs”). The introduction of Mills et al eLife, 2019 (<https://elifesciences.org/articles/48940>; not cited) provides several references that should be mentioned. (Yap et al., 2018; Jain and Vale, 2017; Zhu et al., 2011; McNulty et al., 2017; Johnson et al., 2017; Velazquez Camacho et al., 2017; Shirai et al., 2017; Rošić et al., 2014). The authors should thereby precisely delineate the novelty of their work with respect to this prior work.

We have included the references indicated by Reviewer#2 [see references #30-34] and better cited previous reports demonstrating transcription of STRs. We have also modified the manuscript to better delineate the novelty of our work with respect to these studies: We acknowledge that previous studies reported the transcription of STR-containing RNAs and, as such, STRs can be considered as transcribed. The novelty of our work resides in the discovery that STRs can initiate transcription, therefore not being mere passenger in other RNAs but containing genuine TSSs for distinct RNAs. These clarifications have been included in the Introduction (page 4) and Discussion (page 19) sections and the word ‘initiation’ has been added to ‘transcription’ throughout the revised manuscript.

2. The authors report that 89,948 CAGE initiate at 84,555 STRs. Is that overlap statistically significant? This depends on how overlap is computed. Are the authors using some windows round peaks and if so, how large are they?

Please see our response to Reviewer#1’s comment #1.

To intersect CAGE peaks and STRs, we used the bedtools window and a window of 5 bp upstream and downstream STR coordinates (the exact command line is provided in the Methods section, page 24).

3. The (T)_n class represents the same class than the (A)_n. Is that the reason why these classes have equal sizes in Fig 1B? But then, why do the fractions differ in 1C?

In the HipSTR catalog, (T)_n and (A)_n are distinct classes with 411,609 and 411,236 loci respectively. The y-axis of Figure 1B does not allow to show such small difference. Thus the fractions in Figure 1C differ between (T)_n and (A)_n simply because different loci are considered.

4. I am confused by the fact that the model uses the “3’ end of each STR”. STRs are determined using an algorithm run only on the + strand. How is the 3’ end of an STR defined? What is the biological interpretation of it? Why using the 3’ end for modeling transcription initiation? I would expect focusing on the promoter, i.e. toward the 5’ end and making use of CAGE to define the strand of an STR. More justifications would help.

HipSTR indeed provides a catalog built on the (+) strand but CAGE data are stranded data (see Figure 1A). Thus, CAGE allows to orientate each STR of the HipSTR catalog as exemplified here:

```
** HipSTR catalog (see hg19.hipstr_reference.bed):  
chr1 10001 10468 6 78 Human_STR_1 AACCCCT
```

```
** Same STR with CAGE data (see hg19.hipstr_reference.cage.bed made available at  
https://gite.lirmm.fr/ibc/deepSTR)
```

```
chr1 10001 10468 Human_STR_1;AACCCCT;+ 0.410901 +  
chr1 10001 10468 Human_STR_1;AACCCCT;- 0.354298 -
```

It is then possible to determine the 3’ end of each STR according to the strand considered (here 10468 on the (+) strand and 10002 on the (-) strand).

To build a CNN, we needed aligned sequences with same length. However, as shown in Figure S1, CAGE peaks are scattered along STRs. We thus decided to align the sequences on the 3’ end of the STR, as defined by the CAGE data.

These explanations have now been added in the Methods section of the revised manuscript (page 30).

5. The authors state “the discovery that STR flanking sequences are not inert but rather contain important features that play critical roles in their biology”. This is not exactly novel. Actually, I found the insights of this work to be quite shallow despite previous work on the topic. Notably, Sun et al., Cell, 2018 (although cited by the authors) showed that many disease-associated STRs (daSTRs) are located at boundaries demarcating 3D chromatin domains. Would the DL models improve with chromatin domain annotation? Do the authors find chromatin boundary motifs such as CTCF to be predictive?

As suggested by Reviewer #2, we confronted the work of Sun *et al.*

The figure below shows that daSTRs are associated with high transcription initiation rate as measured by CAGE (enclosed Figure 1, Wilcoxon test p-value = 2.35e-15), confirming our results (new Figure B).

Figure 1 : CAGE signal at all STRs (left) or 26 daSTRs (right) described in Sun *et al.*, Cell 2018. Wilcoxon test p-value = $2.35e-15$

However, we did not observe significant difference in transcription initiation at STRs located

Figure 2 : CAGE signal at STRs located outside (left) or inside (right) TAD boundaries described by Sun *et al.*, Cell 2018 in human cortical plate tissue (Table S4I) and human ES cells (Table S4H). Wilcoxon tests were used to assess statistical difference. p-value = 0.01553 and 0.02017 in human cortical plate tissue and human ES cells respectively.

within or outside TAD boundaries in human ES cells (Table S4H, Wilcoxon test p-value = $0.02017 > 0.01$) and in human cortical plate tissue (Table S4I, Wilcoxon test p-value = $0.01553 > 0.01$)(enclosed Figure 2). Therefore considering the location of STRs in TAD boundaries cannot improve the predictions of our CNNs.

Results obtained interpreting our models (see point #11) did not show enrichment for CTCF motif.

It may be worth noting that Sun *et al.* studied STR flanking regions at a scale much larger than ours. As indicated in Table S1 and in the Method section of their manuscript, ‘the final size of all boundaries is 120kb’. In contrast, we studied 50bp-long STR flanking sequences, making it possible to have several hundreds of STRs within the same TAD boundary. As a consequence, STRs can have different flanking sequences in our case but identical flanking regions according to Sun *et al.*. This major scale discrepancy makes it hard to discuss our work in the light of that of Sun *et al.*.

6. Model performance is not compared to any baseline model. What is the performance of a predictor that returns the median CAGE count for each STR class?

The performances of our models were computed as Spearman correlations. Therefore, computing the performance of the predictor that returns the median CAGE count would require the computation of the correlation between CAGE signal and a constant value. While this is formally not possible, because standard deviation of the vector equals 0 and correlation calculation is not possible, one can consider that such correlation is null.

As a comparison to baseline model, we also computed the correlation between observed CAGE signal and randomized CAGE signal (equivalent to a predictor that returns a random value drawn from observed values). Randomization was repeated 10 times and Spearman correlation is invariably close to 0 (absolute value(ρ) < $5 \cdot 10^{-4}$). This results has been indicated in the Methods section (page 32).

7. Major aspects of the CNN training procedure are not provided. These include: How did the training data was stratified? What are STR class counts in the training and testing data? How many STR's have a CAGE-peak in the training and testing data?

We acknowledge that aspects of the training procedure were initially only provided in the git repository and as a schematic representation in Figure S6. We made an new Supplementary Figure (see new Supplementary Figure S7) to better describe our approach. This figure also includes a table with STR class counts and number of STRs with a CAGE peak in training and testing sets. We also modified the Methods section (pages 30-32) to make the details of the procedure more apparent in the core manuscript.

8. Establishing proper cross-validation in sequence-based predictions can be difficult due to homologous regions. Some of STRS may be part of homologous regions. Deep models may then overfit. The author should assess the issue of homologous regions.

We agree with Reviewer#2 and homologous sequences present in the train and test sets may indeed lead to overfitting. To clarify that point, we used *BLASTn* to look for homology between (*T*)*n* sequences of the test and train sets. The model learned on (*T*)*n* was used because it is the most accurate and therefore the more likely to overfit.

We found 102,209 sequences from the test set with > 60% query cover and identity > 80% with at least one sequence of the train set. We separated these sequences (test set #1, homologous sequences) from the rest of the test set (test set#2, 121,808 non-homologous sequences). We then computed Spearman correlations between the predicted and the observed CAGE signals using these two test sets: 0.73 with test set #1 and 0.78 with test set #2. In both cases, correlations decreases, as compared to that computed on the whole test set (0.84), likely due to differences in CAGE signal distribution between whole test set, test set #1 and #2 (Supplementary Figure S18). However, model performance measured on test set #2 is greater than that obtained with test set #1. This is in contrast to what is expected in case of model overfitting due to sequence homology. We then concluded that homology observed between train and test sets is not sufficient to make the model overfit.

This analysis has been included in the revised version of the manuscript (pages 31-32).

9. The input and target of the CNN should be precisely described. A figure showing how a STR sequence + flanking regions is converted into an input matrix would be very helpful for understanding. Moreover, it is not clear which parts of the sequence exactly were replaced with "N".

We have modified the Figure S6 to better show how STR sequences are converted into one-hot encoded matrices for both classification and regression (see new Supplementary Figure S7). We also provide an example of sequence used in each CNN task at the bottom of the new Figure 5 to show which bases have been replaced by N. The Methods section has also been modified to better explain how and why some bases were replaced by Ns (pages 30, 32 and 33).

10. A density-scatter or hexbin plots showing predicted vs observed CAGE counts on held out data should be provided to visualize the quality of the predictions.

Hexbin plots showing predicted vs observed CAGE signal on held out data have been provided in Supplementary Figure S9.

11. I missed some model interpretation. Can model interpretation technique (see review Eraslan et al. Nat Review Genetics 2019) indicate TF binding sites? For directionality, does the model capture known motifs involved in transcription directionality (e.g. U1 binding sites Almada et al Nature 2013, probably motifs have been reported since then) ?

We sought to identify representations of sequence motifs captured by CNN first layer filters using a strategy inspired by the work of Maslova *et al.* [reference #47]. Details of the analysis have been provided in the revised version of the manuscript (pages 16 and 33). This approach indeed identified several influential first layers correlating with JASPAR PMW scores (results are provided at <https://gite.lirmm.fr/ibc/deepSTR/figures>). However, it is important to remember that our models were optimized to predict CAGE signal, not to learn interpretable representations from input DNA sequences. Koo and Eddy have indeed recently demonstrated that tackling these two questions - prediction and interpretation - requires distinct CNN architectures, in particular adapting max-pooling and convolutional filter size [reference #48]. At present, our models likely learn partial motifs and do not limit the ability to learn motifs in deeper layers. This limitation has clearly been stated in the revised manuscript (page 16).

Regarding transcription directionality, because our CNNs are specifically designed to predict CAGE signal, they cannot learn features or motifs involved in transcription directionality. We nonetheless looked for motifs known to be involved in transcription directionality at canonical TSSs, namely, polyadenylation sites (polyA sites) and U1 binding sites [Almada et al., Nature 2013, reference #40]. These analyses have been included in the revised version of the manuscript (pages 9 and 10). As shown in the new Supplementary Figure S6, we did observe an enrichment of potential U1 binding sites downstream FANTOM CAT TSSs, as previously reported [reference #40], but not downstream $(T)_n$ 3' end. Moreover, while polyA sites are clearly enriched upstream FANTOM CAT TSSs (downstream in the antisense orientation), this observation does not hold true for $(T)_n$ (new Supplementary Figure S6).

Hence, our results suggest that the determinants of transcription directionality at STRs differ from what is observed at canonical TSSs. These results are in agreement with that obtained by Ibrahim *et al.*, [reference #42], who showed that 'a single model of transcription initiation within and across eukaryotic species is not evident.' This conclusion has been added in the revised manuscript (page 10).

12. On page 10, it is stated that 766,747 $(T)_n$ elements are predicted. However, in Figure 1B there are only ~ 400,000 $T(n)$ STR's. Where does this difference stem from?

See also our response to point #4.

Figure 1B shows the number of elements in each STR class according to the HipSTR catalog, which is not stranded. In contrast, the STR sequences used as input in our CNNs are stranded

thanks to the CAGE data, thereby almost doubling the number of elements in each class. This has now been indicated in the Methods section (page 30).

13. Figure 5, 6, and S8 should be provided for metrics computed on held-out test data.

All results depicted Figures 5,6 and S8 were indeed provided with metrics computed on held-out test data, as initially indicated in Figure S6 and the git repository. This has now been also indicated in the core manuscript and in the Methods section (page 30).

14. Page 12. The statement “formally demonstrating the existence of STR class-specific features for transcription prediction.” is too strong. One cannot exclude that one have obtained different results, would other algorithms or mathematical functions have been applied.

We have modified the manuscript to make this statement less strong (see page 13, ‘Overall, the performance of one model tested on another STR class drastically decreases (Figure 5C), revealing the existence of STR class-specific features predictive of transcription initiation.’).

15. Figure 7 should be provided for ClinVar pathogenic and ClinVar benign variants separately (side-by-side) to assess whether there is an enrichment for pathogenic variants. The authors make a strong claim in this direction in the abstract but it is not supported without such comparison.

ClinVar variants considered in Figure 7A are located in a window encompassing STR +/- 50bp (corresponding to the length of sequences used as input in CNN models). In the original Figure S9, we used a window encompassing STR +/- 5bp. We have corrected this mistake and considered a window encompassing STR +/- 50bp in both cases. Initial Figure S9 has now been moved to the core manuscript as Figure 7B.

We performed statistical tests on the results presented in the new Figure 7B. An ANOVA test revealed the existence of differences in CAGE signal at STRs associated with the different classes of ClinVar variants. We next performed pairwise comparisons using Mann-Whitney tests and showed that STRs associated with pathogenic variants exhibit stronger CAGE signal than STRs associated with benign variants (see new Supplementary Figure S12, Wilcoxon test p-value = $1.84e-59$). We therefore modified the sentence ‘The clinical significance of the variants, as defined in the ClinVar database, does not appear directly linked to the transcription rate of STRs’ into ‘Looking at the clinical significance of the variants, as defined in the ClinVar database, we indeed noticed that STRs associated with pathogenic variants exhibit stronger transcription initiation than STRs associated with other variants (Figure 7B and Supplementary Figure S12)’ (page 16).

Second, we looked at the distribution (Supplementary Figure S14A) and the impact (Supplementary Figure S14B) of pathogenic and benign variants. The distribution of both types of variants is similar to that shown in the new Figure 7D and corresponds to the positions identified by our models as key for prediction (new Figure 7C). We did not notice difference in the impact induced by benign and pathogenic SNVs on transcription initiation prediction, as expected provided the results obtained with random and all ClinVar SNVs (new Figure 7C). Thus pathogenic and benign variants are not distinguishable by their distribution around STR 3’end nor their impact on transcription initiation predictions.

We acknowledge that our wording might have been misleading by suggesting a direct link between the distribution of variants around STR 3’end and their clinical impact. We now clearly state (pages 15-17) that (i) pathogenic variants are found at STRs with transcription initiation level higher than that encountered at other STRs affected by other types of variants and (ii) ClinVar variants, whatever their clinical significance, are more frequently found at positions key

for predictions. We re-organized Figure 7 and modified the manuscript, including abstract, accordingly.

Please also see our response to point #18.

16. Fig 7B needs a multiple-testing corrected statistical test to establish that the enrichment shown around the TSS is significant.

A Kolmogorov-Smirnov test was used to compare the distribution of ClinVar variants around STR 3'ends and the distribution of random variations (original Figure 7B). It indicates a statistically significant difference (p -value = $2.95e-11$, indicated page 50 of the revised manuscript).

17. Page 15, this statement is unclear: "Likewise, several diseases were found enriched comparing variant fractions located at transcribed STRs (Fisher's exact test $< 5e-3$, Supplementary Table S2)". What is meant with "variant fraction" and what is meant by "enriched"? I could not reconstruct the Fisher test contingency tables leading to the p -values provided in Suppl. Table S2. Also, multiple testing correction on this large number of tests should be applied.

We have provided all the details of the calculations in the revised Table S2 (first sheet, README). As suggested by Reviewer#2, the p -values were adjusted for multiple testing using the Benjamini and Hochberg correction.

We acknowledge that our wording 'enriched' was misleading since, for some diseases, STRs can be associated with less variants than expected by chance. We have modified the manuscript accordingly (page 16).

18. Page 16 states that the models have learned key positions that correspond precisely to genetic variants linked to human diseases. However, it is stated on page 15: "The clinical significance of the variants, as defined in the ClinVar database, does not appear directly linked to the transcription rate of STRs". How do these statements connect?

As indicated in our response to point #15, we performed new analyses and noticed that STRs associated with pathogenic variants exhibit higher transcription initiation levels than STRs associated with benign variants (new Figure 7B). We also now simply state, in the revised manuscript, that ClinVar variants are more frequently found at positions key for predictions with no difference linked to pathogenicity (page 17). We also conclude that the pathogenicity of ClinVar variants appears to be linked to the transcription initiation level of the targeted STR rather than to the position of the variation or its impact on prediction (page 17).

19. As this claim is provided in the abstract, it should be very precisely nailed or removed.

We have modified the manuscript, including the abstract, to clarify our claims, in accordance to our responses to points #15 and #18 (pages 2 and 15-17).

Minor

20. Some of the other catalogs mentioned on page 5 ("compared to other catalogs.") should be explicitly names and referred to.

We have named the other catalogs (GENCODE, Human BodyMap and miTranscriptome) and provided the references (page 5).

21. Figure 3: To directly compare values in 3A and 3B, a scatter plot of the values in 3B against those in 3A would be more informative (labelling points with STR names). Also, showing confidence intervals (e.g. binomial) would be useful. An alternative to the scatterplot would be side-by-side bar plots (With confidence intervals).

Figure 3A and 3B have been combined into one side-by-side barplot and Binomial proportion 95% confidence intervals have been added (see new Figure 3).

22. Page 9: “as defined in [4],” does not read well. A better style would be “as defined by <name et al> [4],”.

The manuscript has been modified.

23. Page 9, typo: “We used deep” -> “We used a deep”

The manuscript has been corrected.

24. Page 9. It is written that “Transcription at (A)_n, which is mostly detected on the (-) strand, does confirm the observation that transcription at (T)_n is mostly (+).” This sentence suggests that (T)_n is only transcribed on the (+) strand. But what it actually shows is that (T)_n is transcribed in the sense direction of whatever strand it is on, whereas (A)_n is not. That is, the strand where (T)_n is found (the complement of (A)_n) is the one that is transcribed.

We agree with Reviewer#2. Our directionality score computes the ratio between transcription on (+) and (-) strands at STRs, which are systematically defined on the (+) strand in HipSTR catalog i.e. (T)_n on (-) strand are defined as (A)_n. Thus, positive directionality score at (T)_n means that transcription occurs on the (+) strand, where the (T)_n is found. Conversely, negative score detected at (A)_n indicates that transcription occurs on the (-) strand, where (T) is found. We clarified this aspect in the revised manuscript and clearly stated that ‘transcription initiation preferentially occurs on the same strand as (T)_n STRs’ (page 9).

25. Page 10, typo: “approches” -> “approaches”

The manuscript has been corrected.

26. Page 11, typo: “we masked the 7bases located downstream the STR 3’ ends” -> “we masked the 7 bases located downstream of the STR 3’ ends”

The manuscript has been corrected.

27. Page 13. “compare Figure 1B and Figure 6A”. A scatter-plot should be provided to facilitate this comparison, with labelled dots. The R package *ggrepel* offers nice options for dot labelling.

We used the R package *ggrepel* to draw a scatter plot and compare the numbers of loci of each STR class in human and mouse. As in Figure 1B, for sake of clarity, the analysis was restricted to human STR classes with > 2,000 loci. This plot is shown as Supplemental Figure S10 and referenced page 14.

28. Page 13 “Less reliable that the human one” should be “less reliable than the human one” (than)

The manuscript has been corrected.

29. The authors could discuss the observation that STR transcripts mostly reside inside the nucleus and do not get exported.

We have modified the Discussion section (page 20) to state: 'Besides, we show that most CAGE tags initiating at STRs remain nuclear (Figure 4A). This observation suggests that, similar to other repeat-initiating RNAs [references #55,56], STR-initiating RNAs could also play roles at the nuclear/chromatin levels, for instance in DNA topology [references #56,57]'.

However, because we do not provide additional results, we do not feel confident at this stage to go beyond this suggestion.

30. Page 17 "the findings made by by Bertuzzi et al. in": twice "by".

The manuscript has been corrected.

31. Usage of the model: Could the authors describe how the model can be used by practitioners wanting to, for example, interpret variants on patient data. Is there a simple workflow to apply the model to standard bioinformatics files such as variants in a vcf, or must the user manually transform the input data into a model with specific input format?

The models are provided at <https://gite.lirmm.fr/ibc/deepSTR>. They can be used to predict transcription initiation level at STRs using a fasta file. Likewise, impact of genetic variations can be assessed by comparing the predictions obtained for instance with reference and mutated sequences (as in Figure 7). These aspects have been clearly indicated in the Methods section (page 32).

32. Fig 7, "variance". Do you mean "variants"?

We did mean 'variance'. Changes are computed as the difference between these two predictions (reference - mutated, Supplementary Figure S10) and their impact is measured as their variance at each position around STR 3' end (x-axis).

33. The discussion is very repetitive with respect to the introduction.

The Discussion has been modified according to Reviewer#2's comment.

34. On page 26, what does "brut force algorithms" exactly mean? Did the authors try every possible numeric value to find the optimal result? Moreover, the correct spelling is "brute force".

We have provided details of the brute force algorithms used in the methods section (pages 31 and 32) and have corrected the manuscript ('brute' instead of 'brut').

Reviewers' Comments:

Reviewer #1:

Remarks to the Author:

I am satisfied with the responses.

Reviewer #2:

Remarks to the Author:

The authors have addressed all my points.